# SLC13A2 promotes hepatocyte metabolic remodeling and liver regeneration by enhancing de novo cholesterol biosynthesis

Li Shi [1,3], Hao Chen [1,3], Yuxin Zhang [1,3], Donghao An[1], Mengyao Qin[1], Wanting Yu[1], Bin Wen[1], Dandan He[2], Haiping Hao [1,2✉] & Jing Xiong [1✉]

## Abstract

**Metabolic requirements of dividing hepatocytes are prerequisite for liver regeneration after injury. In contrast to transcriptional dynamics during liver repair, its metabolic dependencies remain poorly defined. Here, we screened metabolic genes differentially regulated during liver regeneration, and report that SLC13A2, a transporter for TCA cycle intermediates, is decreased in rapid response to partial hepatectomy in mice and recovered along restoration of liver mass and function. Liver-specific over-expression or depletion of SLC13A2 promoted or attenuated liver regeneration, respectively. SLC13A2 increased cleavage of SREBP2, and expression of cholesterol metabolism genes, including LDLR and HMGCR. Mechanistically, SLC13A2 promotes import of citrate into hepatocytes, serving as building block for ACLY-dependent acetyl-CoA formation and de novo synthesis of cholesterol. In line, the pre-administration of the HMGCR inhibitor lovastatin abolished SLC13A2-mediated liver regeneration. Similarly, ACLY inhibition suppressed SLC13A2-promoted cholesterol synthesis for hepatocellular proliferation and liver regeneration in vivo. In sum, this study demonstrates that citrate transported by SLC13A2 acts as an intermediate metabolite to restore the metabolic homeostasis during liver regeneration, suggesting SLC13A2 as a potential drug target after liver damage.**

**Keywords** TCA Cycle; Partial Hepatectomy; De Novo Cholesterol Synthesis; ATP-citate Lyase; Metabolic Remodeling; Cell Division
**Subject Categories** Metabolism; Molecular Biology of Disease

## Introduction

Liver is constantly exposed to cellular damage which leads to different kinds of liver diseases and contributes as a major cause to global morbidity and mortality (Devarbhavi et al, 2023). Even though some specific therapies exist, host survival and recovery in most cases of liver injury largely rely on the organ's outstanding capability to regenerate (Michalopoulos and Bhushan, 2021). Therefore, mechanistic insights into liver regeneration greatly facilitate the discovery and development of novel therapeutic strategies to improve the management of liver diseases.

Liver regenerative process is conserved in all vertebrates, which is divided into three phases, initiation, progression and termination (Bangru and Kalsotra, 2020). Under normal conditions, 84.1% of hepatocytes lack ability to proliferate, rendering a very low turnover rate. Once the liver is injured from partial hepatectomy or other stimuli, self-renewal or reprogramming of hepatocytes lies in the middle of this integrated organogenesis by which liver regeneration restores not only the original number of hepatocytes but also the full recovery of histological and functional liver tissue (Chen et al, 2020). Dividing hepatocytes secrete many factors that are mitogens for other liver cells, including vascular endothelial growth factor (VEGF) for liver sinusoidal endothelial cells (LSECs), transforming growth factor-α (TGFα) for hepatic stellate cells (HSCs), and granulocyte-macrophage colony-stimulating factor (GM-CSF) for Kupffer cells (Michalopoulos and Bhushan, 2021). Although it is well-known about the transcriptomics regulation that undermines phases of liver regeneration (Fausto et al, 2006), poorly established is the regulation of metabolism and the metabolic requirements of proliferating hepatocytes during the process of liver regeneration (Caldez et al, 2018). Furthermore, metabolic disorders are always linked to the impaired liver regeneration (DeAngelis et al, 2005; Vetelainen et al, 2007), which is a shared characteristic for many liver diseases (Forbes and Newsome, 2016). Therefore, uncovering the metabolic needs of hepatocytes for division during hepatic regeneration has significant clinical implications.

The tricarboxylic acid (TCA) cycle is a central metabolic pathway that oxidizes nutrients to support bioenergetics for cell growth and proliferation (Arnold and Finley, 2023). TCA cycle intermediates are also considered as products of cell metabolism essential for the biosynthesis of macromolecules including lipids, proteins, and nucleotides (Martinez-Reyes and Chandel, 2020). As a typical example, citrate departs from the TCA cycle and is

[1]Department of Pharmacology, School of Pharmacy, China Pharmaceutical University, Nanjing, Jiangsu 210009, China. [2]Jiangsu Provincial Key Laboratory of Drug Metabolism and Pharmacokinetics, State Key Laboratory of Natural Medicines, State Key Laboratory of Natural Medicines, China Pharmaceutical University, Nanjing, Jiangsu 210009, China. [3]These authors contributed equally: Li Shi, Hao Chen, Yuxin Zhang. ✉E-mail: haipinghao@cpu.edu.cn; jxiong@cpu.edu.cn

catalyzed by ATP citrate synthase (ACLY) to secrete acetyl-CoA for the synthesis of fatty acids and cholesterol (Feng et al, 2020), which are not only the structural components for cell plasma membrane but also bioactive lipids. Therefore, TCA cycle is proposed to be important for liver regeneration (Delgado-Coello et al, 2011). TCA cycle activity has to be maintained between anaplerosis and cataplerosis for calibrating anabolic and catabolic pathways (Owen et al, 2002). Besides the enzymes, the metabolic flux of the TCA cycle is also controlled by the efficiency of TCA cycle intermediates transporters, including solute carrier (SLC) family member SLC25A1 and SLC25A10 (Jiang et al, 2016). In the present study, we adopted a classical mouse model of 70% partial hepatectomy for hepatic regeneration to screen the genes which are altered during the phases of initiation and progression using transcriptomics datasets, based on the hypothesis that the hepatocellular membrane SLC transporters play a central role in driving liver regeneration by transporting essential TCA cycle intermediates.

## Results

### SLC13A2 outstands as a downregulated TCA cycle intermediates transporter during the initiation of liver regeneration after partial hepatectomy

As a representative mouse model for liver regeneration after resection of 70% liver mass (PHx), the surgery is frequently applied and extensively studied especially in transcriptomic level to provide insights for the requirements of liver regeneration in vivo (Li et al, 2023b). Therefore, the differentially expressed genes were analyzed using available GEO RNA sequencing (RNA-seq) datasets obtained after well-defined time points during liver regeneration. Available datasets with time points representing the initiation and progression phases of hepatic regeneration (from 20 h to 72 h) after PHx were included to screen out the genes which might undermine liver recovery. The included datasets for screening, GSE95135 (20 h), GSE215423 (24 h), GSE211370 (72 h), GSE188421 (24 h), GSE188768 (24 h), and GSE204901 (48 h) were selected (Fig. 1A). It was notable that *Slc13a2* was the only gene deregulated in all the six GEO datasets, implying a characteristic effect for the initiation and progression of liver regeneration (Fig. 1A,B). When compared with other detectable SLC transporters for TCA cycle associated metabolites, *Slc13a2* showed a typical pattern involving in the process of liver regeneration (Fig. 1C). Different with the ones having similar functions, SLC13A2 decreased at an early stage after PHx surgery and orchestrated the restoration of liver mass and function (Appendix Fig. S1). Since the transported metabolites has extensively revealed roles for generating cellular energy and precursors for biosynthetic pathway (Eniafe and Jiang, 2021), it suggests that SLC13A2 might connect the transcriptomic regulation and the metabolic remodeling required for liver regeneration.

To confirm the association of SLC13A2 with the loss and restoration of liver mass and function during the whole hepatic regenerative process, we also generated a mouse model with partial hepatectomy and analyzed the expression of SLC transporters for TCA cycle intermediates choosing time points representing murine liver regeneration phases of initiation (1 d), progression

(3 d) and termination (7 d) after the surgery (Du et al, 2023). Liver mass and function were lowest at 1 day after the surgery and returned to normal after 7 days, as shown with predictable decrease and recovery of liver/body weight and blood glucose (Fig. 1D,E). Interestingly, the mRNA and protein expression of SLC13A2 were inhibited after 1 day and recovered along with the restoration of liver mass and function, which was also compatible with the increase of cell signaling for liver regeneration, including phospho-AKT, phospho-GSK3β, phospho-JNK, as well as the marker for DNA replication and nuclear division PCNA (Fig. 1F,G). Consistently, in all the SLC transporters for TCA cycle intermediates, *Slc13a2* was also the only one deregulated in the initiation and progression phase of liver regeneration, outstanding from other transporters with similar functions (Fig. 1H).

### SLC13A2 serves as a trigger to drive liver regeneration in the mouse model induced by hepatectomy

We generated hepatocyte-specific SLC13A2 loss- or gain-of-function mice via AAV-mediated transduction and investigated its role in regulating liver physiological functions without PHx. Recombinant AAV8 with a vector expressing sgRNA targeting SLC13A2 was applied to transduce Cas9$^{fl/fl}$ knock-in mice for specific ablation of SLC13A2 in hepatocytes. No consistent differences were observed in the two liver-specific SLC13A2-KO (LKO) groups regarding blood glucose, body weight, liver weight, histological staining and expression of genes involving with glycolysis, TCA cycle and lipogenesis (Appendix Fig. S2A–E). Further, we detected the activation of signaling pathway regulating cell proliferation and found that p-mTOR was slightly decreased while p-GSK3β were significantly increased (Appendix Fig. S2F). Similarly, when we overexpressed SLC13A2 in the liver, body weight, liver weight, blood glucose, histological morphology, gene expression and activation of signaling pathways were all comparable between the groups (Appendix Fig. S3A–F). The data imply that hepatocyte-specific knockout or overexpression of SLC13A2 has limited effects on the liver homeostasis in quiescent liver without the insult of PHx surgery.

To explore the effect of SLC13A2 on liver regeneration, AAV8 with sgRNA targeting SLC13A2 was injected to Cas9$^{fl/fl}$ knock-in mice through tail vein, followed by the surgery of partial hepatectomy. The mice were sacrificed and liver tissues were collected for analysis after different time points indicating the phases of initiation, progression and termination during liver regeneration (Fig. 2A). The ablation efficiency of SLC13A2 was confirmed in RNA and protein level (Fig. 2B,C). The liver/body weight ratio recovery was significantly suppressed with the hepatic deficiency of SLC13A2 (Fig. 2D), which didn't affect the survival rate of mice (Appendix Fig. S4J). As mentioned previously, decrease of blood glucose serves as a powerful initial signal to promote liver regeneration and glucose supplementation suppresses hepatectomy- or hepatotoxin-induced hepatocellular proliferation (Huang and Rudnick, 2014). The blood glucose in SLC13A2-deficient group was increased compared with the control group at 1 d post hepatectomy surgery, indicating a delayed recovery of liver function as well as an impaired signal for the initiation of liver regeneration (Fig. 2E). As hepatocytes with diploid nuclei have distinct metabolic characteristics compared to

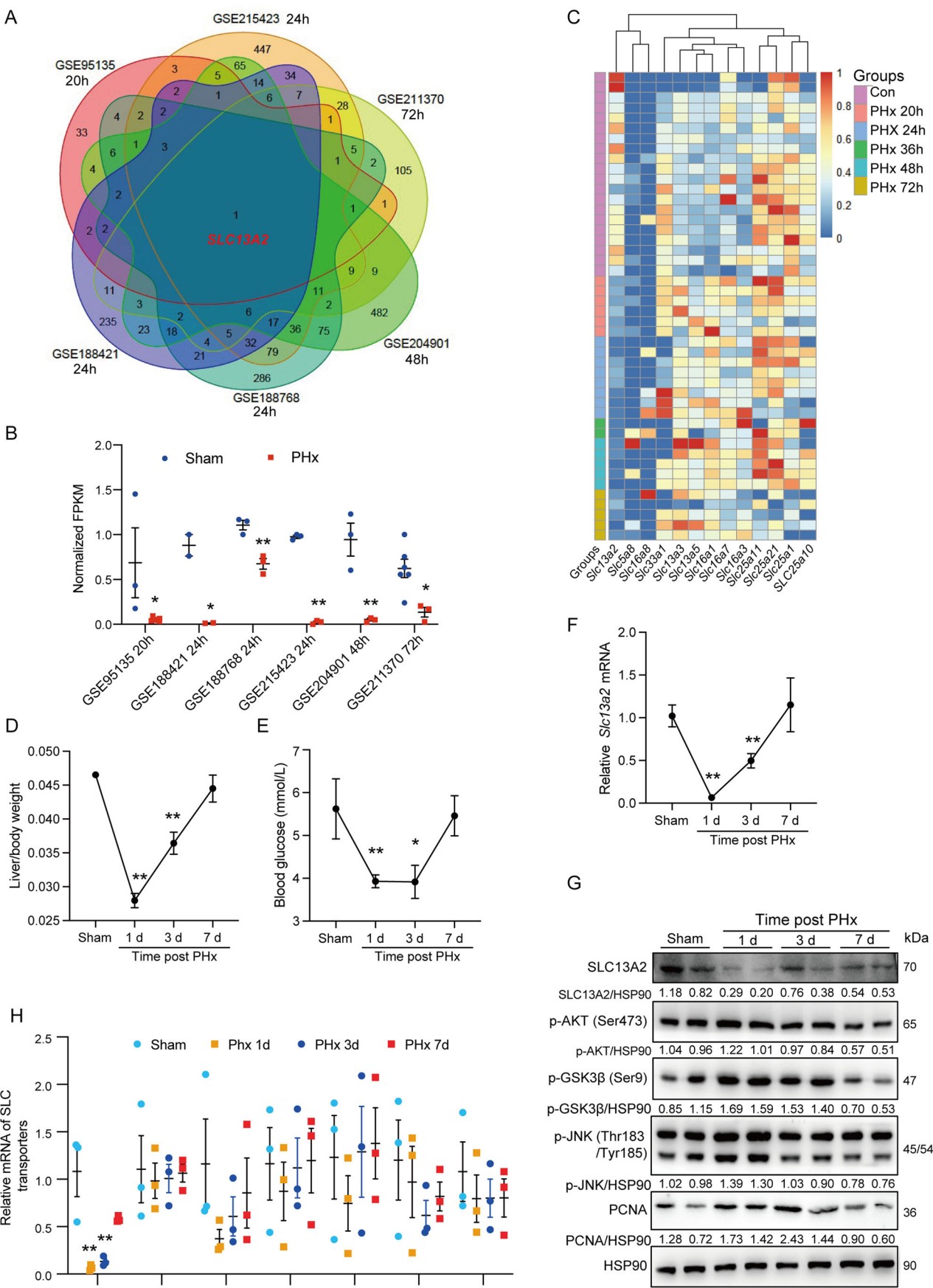

Figure 1.  SLC13A2 is transcriptionally decreased during liver regeneration induced by partial hepatectomy.

(A) Analysis of differentially expressed genes during liver regeneration induced by partial hepatectomy (PHx) in GEO datasets. (B) The expression of *Slc13a2* in the above GEO datasets. GSE95135 ($N = 3$), GSE215423 ($N = 3$), GSE211370 (Con, $N = 6$; PHx, $N = 3$), GSE188421 ($N = 2$), GSE188768 ($N = 3$) and GSE204901 ($N = 3$). (C) Heatmap showing the expression of detectable solute carrier transporters for glucolipid metabolism intermediates. GSE95135 (Con, 20 h), GSE215423 (Con, 24 h), GSE211370 (Con, 72 h), GSE188421 (Con, 24 h, 36 h, 48 h, 72 h), GSE188768 (Con, 24 h) and GSE204901 (Con, 48 h) were selected. Gene expression data from these datasets were converted to TPM and then merged. Batch effects were removed using the Combat function from the sva package. Subsequently, all data were scaled within genes using Min-Max Normalization to the range [0,1] to facilitate display in a single graph. (D–F) Liver weight recovery (D), blood glucose (E) and RNA expression level of *Slc13a2* in different time points after the surgery of PHx ($N = 5$ for sham, $N = 7$ for PHx 1 d and 3 d and $N = 5$ for PHx 7 d). (G) Protein expression level of SLC13A2 and hepatocellular proliferating signaling pathway in indicated time points after PHx surgery. (H) RNA expression level of SLC transporters for TCA cycle intermediates ($N = 3$). Data are expressed as Mean ± SEM. *$P < 0.05$ and **$P < 0.01$ PHx vs Sham group, two-tailed unpaired Student's $t$ test (B, H), one-way ANOVA followed by Bonferroni multiple comparisons (D–F). Source data are available online for this figure.

cells with polyploid which significantly increases with cell proliferation during liver regeneration (Kreutz et al, 2017; Yagi et al, 2020), the histological staining delineated less abundant double-nucleated hepatocytes with polyploid in the knockout group compared with control, suggesting halted hepatocellular proliferation for liver regeneration (Fig. 2F). Consistently, we performed immunohistochemical staining of Ki67, a marker for DNA replication and cell proliferation, and found that the ratio of Ki67-positive cells was significantly less with SLC13A2 deficiency in hepatocytes (Fig. 2G,H).

Following, a mouse model was generated with hepatocyte-specific overexpression of SLC13A2 using AAV8 with TBG promoter. Partial hepatectomy was performed to induce liver regeneration after 2 weeks of AAV injection and the mice were sacrificed for analysis at designed time points after the surgery (Fig. 3A). The overexpression efficiency of SLC13A2 in the liver was verified at the mRNA and protein levels (Fig. 3B,C). Liver/body weight ratio was significantly increased with unchanged survival rate, while blood glucose at 1 day after the surgery was decreased with the overexpression of SLC13A2, indicating accelerated restoration of liver mass and functions by SLC13A2 (Fig. 3D,E; Appendix Fig. S4D). Histological staining revealed that hepatocytes with double nuclei were increased in the SLC13A2 overexpression group compared to the control group, suggesting that more hepatocytes gained the capability to proliferate (Fig. 3F). Consistent with the increased percentage of polyploid hepatocytes, immuno-histochemical staining of Ki67 also showed increased nuclear positivity in the SLC13A2 overexpression group during liver regeneration (Fig. 3G,H).

Given that the activation of cell cycle and DNA replication are important in stimulating robust regenerative hepatocellular proliferation and that changes in the transcriptome return to the initial state after 72 h post hepatectomy (Li et al, 2023b), we analyzed the expression of genes controlling cell cycle progression and DNA replication, including *Ccnd1*, *Ccnb1*, *Mcm2*, *Cdc45*, and *Foxm1*, after 1 d and 3 d post PHx, to further confirm the role of SLC13A2 on driving regenerative hepatocellular proliferation during PHx-induced liver regeneration. It was demonstrated that SLC13A2 deficiency inhibited the expression of these checkpoint components of cell cycle machinery *Ccnd1*, *Ccnb1* and *Cdc45*, transcription factor for the expression of cell cycle genes *Foxm1*, as well as a well-known DNA replication licensing factor *Mcm2* (Fig. 4A up). In the contrast, hepatic overexpression of SLC13A2 increased the expression of these proliferative genes compared to the control group, including *Ccnd1*, *Cdc45*, *Mcm2*, and *Foxm1* (Fig. 4A bottom). As intracellular signals are activated within

hepatocytes after hepatectomy to promote liver regeneration, we detected the activation of JNK and ERK (which is activated by EGFR ligands and HGF) and phosphorylation of GSK3$\beta$ (which leads to CCND1 accumulation and accelerates hepatocyte proliferation (Lai et al, 2016)). It showed that the phosphorylation of JNK, ERK and GSK3$\beta$ were all suppressed with the knockout, and elevated with the overexpression of SLC13A2 at 1 day and 3 days after the surgery of hepatectomy (Fig. 4B). Considering the dynamic alteration of SLC13A2 expression in response to partial hepatectomy (Fig. 1F,G), these data further suggest that SLC13A2 responds quickly to the metabolic change bottoming out at 24 h after the hepatectomy surgery, and the elevation of SLC13A2 during the progression phase ensures the capability of hepatocyte to divide for the restoration of the original liver mass. The results demonstrate that SLC13A2 increases the capacity of hepatocyte to reenter the cell cycle and plays a positive role to accelerate hepatocellular proliferation for liver regeneration after partial hepatectomy.

## Cholesterol metabolism represents metabolic remodeling stimulated by SLC13A2 during hepatic regeneration

As a metabolite transporter, it is intriguing to undermine the mechanisms of SLC13A2 in promoting liver regeneration focusing on metabolic regulation. Firstly, we screened metabolites which was increased by SLC13A2 in primary hepatocytes using untargeted metabolomics analysis. The significant increase of metabolites in nucleotide metabolism induced by SLC13A2 indicates that this transporter remodeled the metabolic state for hepatocellular proliferation and that primary hepatocytes could be a reliable cell model to reflect the hepatocytes metabolic remodeling during liver regeneration (Fig. 5A). Secondly, we evaluated the gene expression for metabolic regulation, including glucose and lipid metabolism, and found the genes for glycolysis, TCA cycle, fatty acid synthesis and transportation, including *Fasn*, *Srebp1c*, *Hk2*, *Idh2*, *Fabp4*, were unchanged with the manipulation of SLC13A2 (Appendix Fig. S5A,B). It was noted that genes for de novo cholesterol synthesis, including *Srebp2*, *Ldlr*, *Hmgcr*, *Fdft1*, were consistently increased in SLC13A2 overexpression group, and decreased in the deficiency group (Fig. 5B). Compatibly, we found that the expression of ACC1, FASN and activation of SREBP1 were not changed during the process of liver regeneration, while the expression of LDLR and activation of SREBP2 were significantly increased at 1 day after hepatectomy (Appendix Fig. S5C). It echoes with previous statement that regenerative liver uptakes

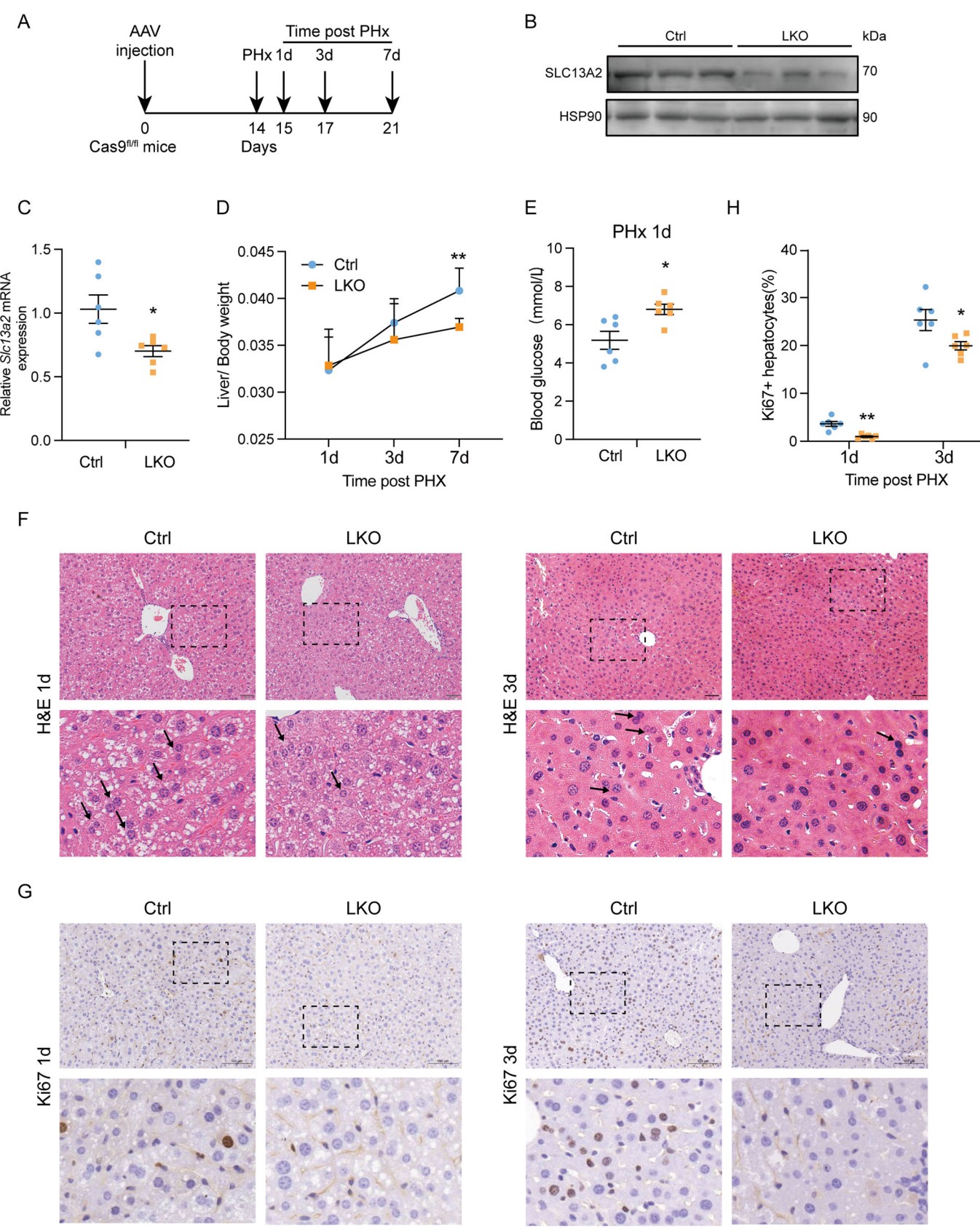

◄ **Figure 2. Liver-specific knockout of SLC13A2 suppresses liver regeneration induced by partial hepatectomy.**

(A) Diagrams of liver-specific knockout (LKO) of SLC13A2 and study design. AAV-TBG-Cre-U6-Lacz (Ctrl) and AAV-TBG-Cre-U6-sgRNA (LKO) virus ($5 \times 10^{11}$ VP/mouse) were injected to the Rosa26-FSF-Cas9 knockin mouse via tail vein. The transduced mice were subjected to the PHx surgery. (B) Protein expression level of SLC13A2 with AAV-transduced liver-specific knockout. (C) RNA expression level of *Slc13a2* in whole liver lysates. (D) Liver weight recovery in different time points after the surgery of PHx. (E) Blood glucose at 1 day after the surgery of PHx. (F, G) Representative images of H&E staining (F, scale bar, 50 µm) and Ki67 IHC staining (G, scale bar, 100 µm). Arrow heads are pointing to the double-nucleated hepatocytes. (H) Quantification for the ratios of Ki67-positive cells. Data are expressed as Mean ± SEM ($N = 6$). *$P < 0.05$ and **$P < 0.01$ PHx vs Sham group, two-tailed unpaired Student's $t$ test for analysis between two groups (C, E), two-way ANOVA followed by Bonferroni multiple comparisons (D, H). Source data are available online for this figure.

adipose-derived fat store through hypoglycemia-induced adipose lipolysis, rather than hepatic de novo fatty acid synthesis for the transient liver steatosis from 12 to 24 h after the surgery, which is required for liver regeneration in mice (Newberry et al, 2008).

To verify the effects of SLC13A2 on regulating cholesterol instead of fatty acid synthesis, we evaluated the content of lipids using lipidomics analysis along with fatty acids and cholesterol standards after overexpression in primary hepatocytes in the presence or absence of EGF to mimic the essential humoral regulators for hepatocellular proliferation encountered by hepatocytes during liver regeneration (Kang et al, 2012). The results revealed an elevation of cholesterol and its synthesis metabolite Zymostenol (5a-cholest-8-en-3b-ol) induced by the treatment of EGF or SLC13A2 respectively. Also, this induction by SLC13A2 persisted no matter in the presence or absence of the EGF treatment, which was distinct with the pattern of triglyceride and other fatty acid derivatives (Fig. 5C). Then, we analyzed the maturation of SREBP2 as well as the protein expression of LDLR and HMGCR in response to SLC13A2 expression during in vivo liver regeneration. The results further revealed that these genes for cholesterol synthesis and transportation were regulated by SLC13A2 to stimulate hepatocellular proliferation after the mice were subjected to partial hepatectomy (Fig. 5D). Echoed with the unchanged expression during liver regeneration process, maturation of SREBP1 and the expression of lipogenic proteins, ACC1 and FASN, were also unchanged with the deletion of SLC13A2 (Appendix Fig. S5D). It supports that cholesterol metabolism represents as a typical characteristic of metabolic remodeling induced by SLC13A2 during liver regeneration in response to partial hepatectomy.

It is inspiring to know whether the promotion of liver regeneration induced by SLC13A2 is hepatectomy specific, therefore, we established a chronic liver injury model by feeding HFD for 20 weeks as described (Fan et al, 2023) in the mice of SLC13A2 liver-specific deficiency (Fig. 6A). There was no significant difference in body weight, blood glucose, liver weight, adipose tissue weight, glucose and insulin tolerance between the two groups. Interestingly, serum AST activity was increased significantly in LKO mice, indicating deteriorated liver injury (Fig. 6B). In mice with SLC13A2 disruption, serum TC raised (Fig. 6C), while liver TG and TC content declined (Fig. 6D), in accordance with the suppressed expression of LDLR (Fig. 6I), a receptor for serum cholesterol transportation to the liver. Akin to the effects of SLC13A2 on liver regeneration induced by PHx, the number of hepatocytes with double nuclei shrank (Fig. 6E), and the percentage of cells with nuclear Ki67 positivity also reduced (Fig. 6F,G) when liver specifically deleted with SLC13A2. More intriguingly, sirius red staining revealed increased collagen fibers in the mouse liver

with SLC13A2 deficiency (Fig. 6H), indicating suppressed regenerative capacity resulted in hepatocyte injury and hepatic fibrosis, especially under chronic damaging conditions (Haideri et al, 2017; Li et al, 2023a). Also, the maturation of SREBP2, expression of LDLR and HMGCR, as well as the signaling for hepatocellular proliferation, phosphorylation of JNK, ERK and GSK3$\beta$, were all suppressed by SLC13A2 deficiency (Fig. 6I). The data prove that SLC13A2 also enhances liver regeneration in HFD-induced chronic liver injury by multiplying cholesterol synthesis for hepatocellular proliferation.

## De novo cholesterol synthesis endorses the hepatic regenerative capability promoted by SLC13A2

To investigate the dependence of SLC13A2-accelerated liver regeneration on de novo cholesterol synthesis, AAV8 (GFP vs SLC13A2) were injected into the mouse tail vein to induce hepatic overexpression of SLC13A2. Following, HMGCR inhibitor lovastatin was given for consecutive 3 days prior to the partial hepatectomy to inhibit the promoted cholesterol synthesis required for accelerated liver regeneration by SLC13A2. Three more days after the surgery, the mice were injected with BrdU to detect proliferating hepatocytes at 3 h before the sacrifice for analysis (Fig. 7A). Again, the overexpression of SLC13A2 enhanced liver regeneration as liver/body weight ratio was increased reaching a statistical significance at 3 d post the surgery, which was consistent with previous findings (Figs. 7B and 3D). Moreover, serum and liver TC were all elevated with the overexpression of SLC13A2, while serum TG also significantly increased and liver TG had a tendency to increase by SLC13A2. Interestingly, the treatment of lovastatin potently decreased the level of serum and liver TC (Fig. 7C), along with the suppressed liver mass restoration in SLC13A2 overexpression group (Fig. 7B).

Compatibly, BrdU incorporation in hepatocellular nuclei representing DNA replication was also boosted by SLC13A2, which was significantly inhibited by the pretreatment of cholesterol synthesis inhibitor lovastatin (Fig. 7D,E; Appendix Fig. S6F). The increase of blood glucose reflecting the restoration of liver function was also induced by SLC13A2, which was suppressed by the pretreatment of lovastatin (Fig. 7F). Mechanistically, lovastatin abolished the effects of SLC13A2 on the expression of genes for cholesterol synthesis *Srebp2* and *Hmgcr*, as well as proliferating genes *Foxm1* and *Mcm2* (Fig. 7G). Furthermore, SLC13A2-induced maturation of SREBP2, increased expression of LDLR and HMGCR, elevated activation of proliferating signaling including p-ERK, p-JNK, p-AKT, and p-GSK3$\beta$, and promoted expression of proliferative gene PCNA were all attenuated by the treatment of

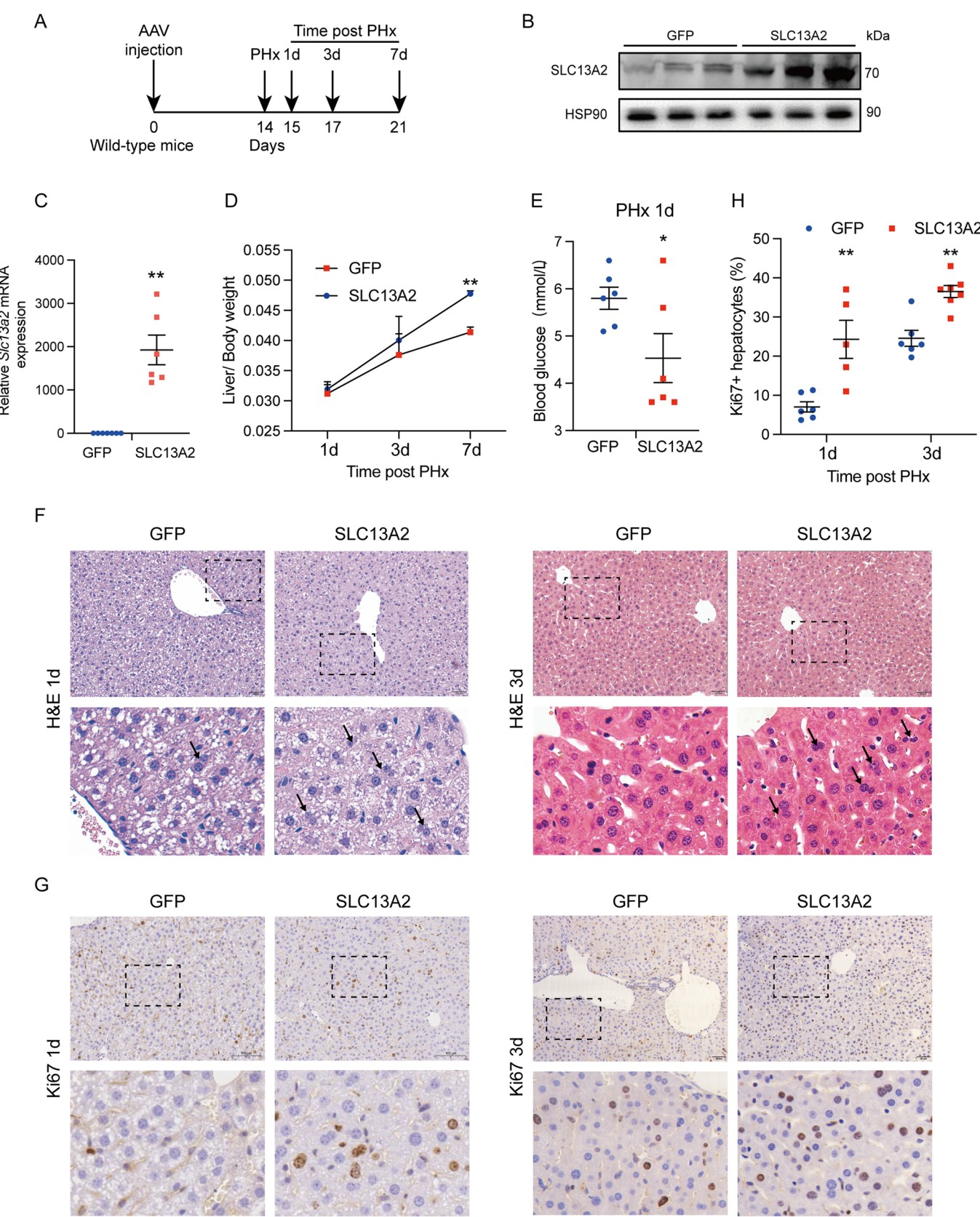

**Figure 3. Liver-specific overexpression of SLC13A2 promotes liver regeneration induced by partial hepatectomy.**

(A) Diagrams of liver-specific overexpression of SLC13A2 and study design. AAV-TBG-GFP (GFP) and AAV-TBG-SLC13A2 (SLC13A2) virus (1 × 10^11 VP/mouse), were injected to the wild-type mouse via tail vein. The transduced mice were subjected to the PHx surgery. (B) Protein expression level of SLC13A2 with AAV-transduced liver-specific overexpression. (C) RNA expression level of *Slc13a2* in whole liver lysates. (D) Liver weight recovery in different time points after the surgery of PHx. (E) Blood glucose at 1 day after the surgery of PHx. (F, G) Representative images of H&E staining (F, scale bar, 50 μm) and Ki67 IHC staining (G, scale bar, 100 μm). Arrow heads are pointing to the double-nucleated hepatocytes. (H) Quantification for the ratios of Ki67-positive cells. Data are expressed as Mean ± SEM (*N* = 6). *$P < 0.05$ and **$P < 0.01$ PHx vs Sham group, two-tailed unpaired Student's *t* test for analysis between two groups (C, E), two-way ANOVA followed by Bonferroni multiple comparisons (D, H). Source data are available online for this figure.

lovastatin (Fig. 7H). The above data demonstrates that SLC13A2-induced regeneration capacity in the liver is dependent upon cholesterol synthesis.

In order to establish a reliable cell model for better investigating the mechanisms of SLC13A2-stimulated regenerative hepatocellular proliferation, mouse normal hepatocyte cell line AML12 cells and primary mouse hepatocytes were infected with adenovirus for the overexpression of SLC13A2 in vitro, as the two types of cells shared complementary advantages. Cell viability assay showed that gain-of-function effects of SLC13A2 increased the proliferating rate of AML12 cells, which was confirmed by the colony formation analysis with increased colony numbers and areas induced by SLC13A2 (Fig. 8A). Then, we used filipin fluorescence to stain cholesterol in AML12 cells and primary hepatocytes with the overexpression of SLC13A2. Both cells exhibited increased accumulation of cholesterol in the cells, as primary hepatocytes mostly trapped cholesterol in the cytosol while AML12 cells mainly spread cholesterol to the cell membrane probably due to its rapid proliferative rate (Fig. 8B). Therefore, we isolated and cultured primary hepatocytes from the AAV-sgRNA injected LKO mice and detected the proliferating signaling and proteins for hepatocellular proliferation and cholesterol synthesis. It was revealed that the deletion of SLC13A2 inhibited the activation of ERK, GSK3β and JNK, as well as the expression of PCNA, HMGCR, LDLR and maturation of SREBP2. We treated these cells with cholesterol (50 μM) and found that all the signaling and gene expression suppressed by SLC13A2 deletion was almost fully rescued by the treatment of cholesterol (Fig. 8C left). Similarly, when we overexpressed SLC13A2 using adenovirus in primary hepatocytes, the stimulating effects of SLC13A2 on proliferation and cholesterol synthesis were nearly completely ablated by the treatment of lovastatin (10 μM) in vitro (Fig. 8C right). These data further prove the important roles of cholesterol synthesis in SLC13A2-improved liver regeneration.

## SLC13A2 transports citrate which is cleaved by ACLY to secrete acetyl-CoA for cholesterol synthesis during liver regeneration

In order to elucidate the mechanisms how SLC13A2 augmented cholesterol synthesis, we isolated cell fractions including nuclei, mitochondria, cytosol and plasma membrane in primary hepatocytes with the overexpression of SLC13A2 and found that SLC13A2 was expressed on the plasma membrane instead of other intracellular compartments (Fig. 8D), supporting that it acts as a membrane transporter to import some essential metabolites for cell proliferation. Then, we used UPLC-MS-MS to detect the

relative content of the putative substrate metabolites which was reported to be transported by SLC13A2 (Bergeron et al, 2013). In the liver tissues at 1 day after partial hepatectomy, the content of citrate was significantly increased by SLC13A2 to about 20-fold higher, while the other putative substrates including succinate, malate and 2-oxoglutarate (α-ketoglutarate) were not significantly changed (Fig. 8E). It implies that it is citrate that is transported by SLC13A2 to liver during liver regeneration. Then, we detected citrate abundance in the extracellular and intracellular compartment of primary hepatocytes and found that SLC13A2 caused a potent reduction of extracellular citrate, but no difference of intracellular citrate, indicating that SLC13A2-imported citrate was converted into other metabolites rapidly in the cells (Fig. 8F). Further, targeted metabolomic assay using 13C-labeled citrate also revealed more percentage of labeled citrate in SLC13A2-overexpressed cells (Fig. 8G). It confirms that it is citrate that is transported by SLC13A2 into the hepatocytes.

To verify the mechanisms of SLC13A2 in the regulation of cholesterol synthesis genes in a citrate-dependent manner, we applied citrate (200 μM) to primary hepatocytes with or without the deficiency of SLC13A2. It was observed that the upregulation of cholesterol synthesis genes, *Srebp2*, *Hmgcr*, *Ldlr*, and *Fdft1*, protein levels of SREBP2 and HMGCR, and that the activation of cellular signaling, p-GSK3β and p-ERK, were resulted from the treatment of citrate. Interestingly, the effects of citrate were totally abolished by the depletion of SLC13A2 in hepatocytes (Fig. 8H,I). The results support the conclusion that SLC13A2 transports citrate to the hepatocytes for the synthesis of cholesterol which facilitates hepatic regeneration.

Given that intracellular citrate is catalyzed by ACLY to produce oxaloacetate and acetyl-CoA which can be the building blocks for the synthesis of cholesterol (Chen et al, 2024; Feng et al, 2020), we applied ACLY inhibitor (ACLYi, BMS-303141, TargetMol, USA) to mice with SLC13A2 overexpression to reveal the dependence of this transporter on citrate transportation for liver regeneration in vivo (Fig. 9A). Again, SLC13A2 significantly increased liver/body weight ratio, blood glucose and liver TG/TC content, which was potently attenuated by the treatment of ACLYi (Fig. 9B–E). In addition, ALCYi treatment also suppressed the SLC13A2-promoted gene transcriptional expression for cholesterol synthesis and cell proliferation, including *Srebp2*, *Ldlr*, *Mcm2*, and *Cdc45* (Fig. 9F). BrdU staining, gold standard for liver regeneration, also showed decreased nuclear positivity by ACLYi treatment in mice with SLC13A2 overexpression (Fig. 9G). Following, ACLYi was given to the SLC13A2-overexpressed primary hepatocytes, then the expression of proliferating and cholesterol-synthesis genes, and activation of cell signaling were evaluated. It was illustrated that the

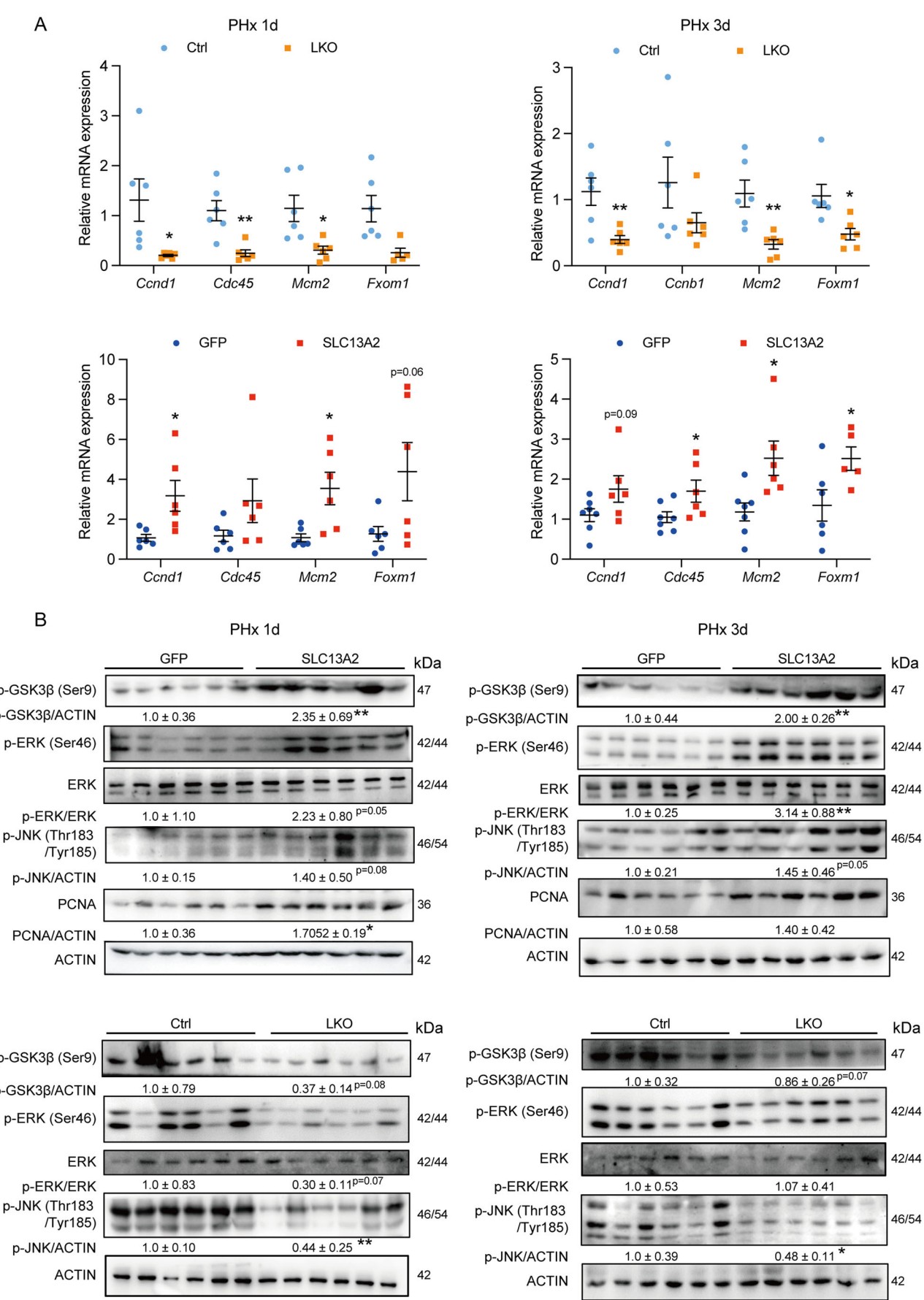

**Figure 4. SLC13A2 regulates gene expression and activation of signaling for cell proliferation during liver regeneration induced by partial hepatectomy.**

(A) RNA expression of genes involving in cell proliferation. (B) Activation of signaling pathway in cell proliferation during PHx-induced liver regeneration. Data are expressed as Mean ± SEM ($N = 6$). *$P < 0.05$ and **$P < 0.01$ SLC13A2 vs GFP group, or LKO vs Ctrl group, two-tailed unpaired Student's $t$ test. Source data are available online for this figure.

application of ACLYi (15, 20 and 25 μM) inhibited the activation of p-AKT, p-GSK3$\beta$, p-ERK, and p-JNK, repressed the expression of PCNA, LDLR and HMGCR, as well as the maturation of SREBP2 in a concentration-dependent manner in hepatocytes with SLC13A2 overexpression (Fig. 9H). Further, we treated the SLC13A2-overexpressed AML12 cell with ACLYi (25 μM) and determined the rate of EdU incorporation into the hepatocellular nuclei. We found that the treatment of ACLYi decreased the ratio of EdU-positive cells, which was rescued by the application of cholesterol (50 μM) in SLC-overexpressed cells (Fig. 9I). Similar, ACLYi (25 μM) suppressed the cholesterol synthesis in SLC13A2-overexpressed cells, which was also reversed by cholesterol treatment (50 μM) (Fig. 9J). Taken together, we establish a novel model that SLC13A2-transported citrate is catalyzed by ACLY to secrete acetyl-CoA for the synthesis of cholesterol, which acts as building blocks for hepatocellular proliferation during liver regeneration (as shown in the synopsis).

## Discussion

The morphological and functional loss of liver due to injury or surgical resection initiates liver regeneration to ensure the stability of liver function and size (Groeneveld et al, 2019; Michalopoulos and Bhushan, 2021). Regeneration impotence leading to liver failure or even death more frequently happens in patients with metabolic disorders (Mendes-Braz and Martins, 2018). Even though the transcriptional remodeling is extensively explored through transcriptomic analysis to provide some perspective into the mechanisms for liver regeneration, how transcriptional regulation is translated into metabolic compatibility is still unclear. The present study proposes a model to define the metabolites transportation mediated by SLC transporter which connects the transcriptomic regulation and metabolic remodeling required for liver regeneration after partial hepatectomy (as shown in synopsis). A better understanding of the metabolic switch that promotes hepatocyte division during liver regeneration will help to identify potential therapeutic targets to improve patient prognosis with liver diseases (Solhi et al, 2021).

Liver regeneration has huge energy demand, therefore, glucose and lipid metabolism adapt accordingly to meet the energetic requirements (Solhi et al, 2021). Hepatic lipid accumulation which peaks in 12–24 h after liver hepatectomy has been demonstrated to be essential for normal liver regeneration (Chen et al, 2023). Solute carrier family (Solute Carrier Family, SLCs) transporters represent a large family associated with a variety of metabolic diseases and are abundantly expressed in the liver (Zhang et al, 2019). They transport nutrients or TCA cycle metabolic intermediates, regulating nutrient supply, metabolic transformation, energy balance and oxidative stress, and thus play a key role in the regulation of liver homeostasis. Deep knowing about how

SLC transporters are transcriptionally regulated during the process of liver regeneration will provide insights into the mechanisms of hepatocyte getting energy support for liver restoration after injury or liver resection. It is interesting that SLC13A2 is the only one deregulated SLC transporter in the initiation and progression phases of liver regeneration shared by the available GEO datasets (Fig. 1A). Since some SLC transporters have been successfully developed as drug targets, for example dapagliflozin, an inhibitor of sodium-glucose cotransporter 2 (SGLT2/SLC5A2) for the treatment of type 2 diabetes (Hanefeld and Forst, 2010), SLC transporters as a group are considered promising drug targets of great significance for drug discovery and development, only secondary to G protein coupled receptors (Lin et al, 2015). Therefore, our findings provide perspective into a potential drug target to facilitate liver regeneration after hepatectomy.

SLC13A2 belongs to SLC13 family which encode multiple transmembrane transporters including SLC13A1 ~ 5 (Chi et al, 2024). Among them, SLC13A2 (sodium di- and tri-carboxylate cotransporters 1, NaDC1), SLC13A3 (sodium di- and tri-carboxylate cotransporters 3, NaDC3) and SLC13A5 (sodium-coupled citrate transporter, NaCT) share similar transporting substrates, mainly TCA cycle metabolites as succinate, citrate and α-ketoglutarate (Markovich and Murer, 2004). Both SLC13A3 and SLC13A5 are not changed during the process of liver regeneration, indicating irrelevant role for the metabolic remodeling in this process (Fig. 1C,H). In all the substrates, SLC13A2 has highest affinity for succinate which is followed by citrate (Km 0.35 mM for succinate and 0.6 mM for citrate) (Colas et al, 2015). Nevertheless, the concentration of citrate in blood stream of healthy people is higher than that of succinate, which facilitates SLC13A2 to transport citrate rather than succinate into the liver (Fig. 8E) (Costello and Franklin, 2016; Serena et al, 2018). It is demonstrated that SLC13A2 expresses on the hepatocyte plasma membrane, acting as a transporter for the import of citrate (Fig. 8D–G). The transported citrate subsequently acts as an essential material for the synthesis of cholesterol (Fig. 8H,I) which is important for hepatocellular proliferation as both structural blocks and active signals during liver regeneration (Delgado-Coello et al, 2011). Increased acetyl-CoA production stimulates lipid synthesis, including activation of SREBP2 and promoted expression of HMGCR and LDLR (Seo et al, 2019). Acetyl-CoA may also increase chromatin acetylation at cholesterol-biosynthesis gene loci to elevate the protein stabilization or expression of these genes (Cai et al, 2019; Giandomenico et al, 2003). The previous findings support our conclusion that SLC13A2 upregulates cholesterol synthesis gene expression by providing essential precursor metabolite acetyl-CoA which is released from citrate through the cleavage by the enzyme ACLY (Fig. 9). It is interesting that the application of cholesterol to primary hepatocytes with SLC13A2 deficiency increases SREBPs cleavage, N-SREBP2 and the

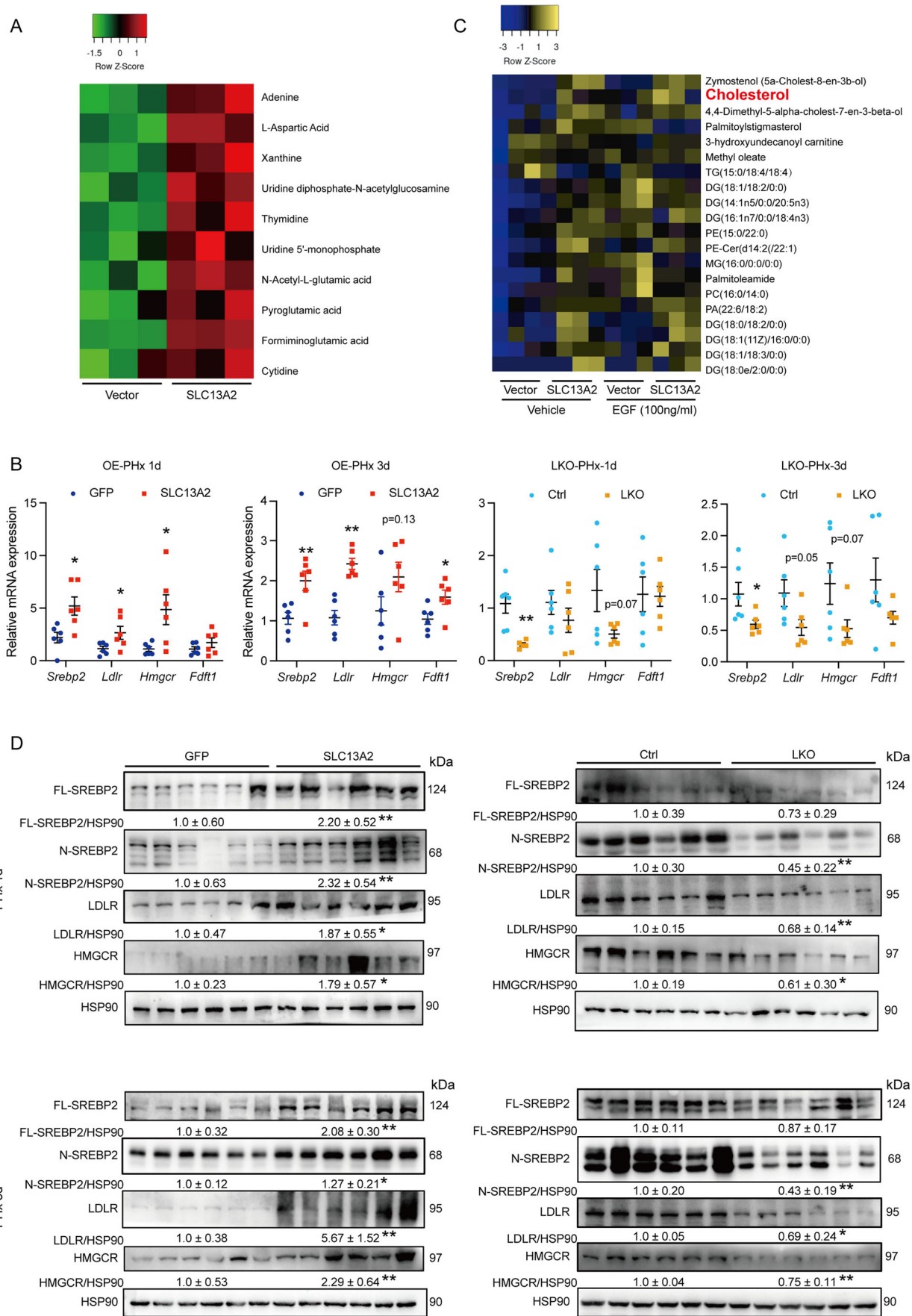

**Figure 5. SLC13A2 promotes cholesterol metabolism in PHx-induced liver regeneration.**

(A) Primary hepatocytes were transfected with vector or SLC13A2 adenovirus and cultured for additional 60 h. The metabolites were analyzed using untargeted metabolomics method ($N = 3$). (B) RNA expression of genes involving in cholesterol synthesis at 1 day and 3 days after PHx ($N = 6$). (C) Primary hepatocytes were transfected with vector or SLC13A2 adenovirus and cultured for additional 48 h, following with the treatment of EGF (100 ng/mL) for 24 h. The metabolites were analyzed using untargeted lipidomics method ($N = 3$). (D) Immunoblots of protein expression in cholesterol synthesis in the PHx model. Data are expressed as Mean ± SEM. *$P < 0.05$ and **$P < 0.01$ SLC13A2 vs GFP group, or LKO vs Ctrl group, two-tailed unpaired Student's $t$ test. Source data are available online for this figure.

expression of HMGCR and LDLR (Fig. 8C left). In a setting with SLC13A2 deficiency, cholesterol synthesis is suppressed (Figs. 5B,D and 8C left). When cholesterol abundance is low, mTORC1 localizes in the cytosol and remains inactive. When intracellular cholesterol concentration increases, mTORC1 translocates to the lysosomal membrane and activates its kinase function, which initiates anabolic pathway for cell growth (Lim et al, 2019; Xie and Bernt, 2022). Activation of mTORC1 triggers SREBP2 cleavage and translocation (Eid et al, 2017), phosphorylates USP20 and subsequently stabilizes HMGCR at the fasting-feeding transition (Lu et al, 2020). This fasting condition reflects insufficient cholesterol supply, while refeeding means regained supply of cholesterol, which is very consistent with our conditions. Our findings are distinct with previous dogma that cholesterol inhibits SREBP2 hydrolysis and transcription of HMGCR in a feed-back manner. A recent study using integrated transcriptomic and metabolomic analyses also reveals that disturbances of cholesterol metabolism pathway underlies the delayed liver regeneration in extensive hepatectomy compared with partial hepatectomy (Li et al, 2023b). It confirms that cholesterol metabolism plays an important role in the efficient liver regeneration after partial hepatectomy.

Regarding the possible reasons for the downregulation of SLC13A2 in the initiation phase of liver regeneration (Fig. 1A,F,H), the complicated insults in the early stage of PHx surgery should be dissected to possibly give a satisfying answer. For example, glucose deprivation, increased fatty acid and inflammatory cytokines which are encountered by the hepatic cells during this stage should be considered. Some transcriptomics datasets have provided the lead as HFD feeding suppresses the expression of SLC13A2 significantly (GSE68360). In addition, several transcription factors might get involved in the transcriptional regulation of SLC13A2 during liver regeneration, including HNF4$\alpha$ (GSE210842), which is responded to signals of fatty acid and inflammation. Previous study reveals a rapid decline in nuclear and cytoplasmic HNF4$\alpha$ protein levels, along with decreased target gene expression, within 1 h after PHx and recovered after 3 d. Also, hepatocyte-specific deletion of HNF4$\alpha$ results in 100% mortality of this surgery (Huck et al, 2019). What's the role of SLC13A2 in HNF4$\alpha$-involved maintenance of liver regenerative capacity deserves full address in the future. Further elucidation on the regulatory mechanisms of SLC13A2 expression in response to the liver injury signals will provide insightful suggestions for drug discovery strategies targeting this transporter.

It is noted that the blood glucose regulated by SLC13A2 is different after 1 d and 3 d of PHx surgery. At 1 d post PHx, overexpression of SLC13A2 repeatedly decreases blood glucose (Fig. 3E). While at 3 d post PHx, the blood glucose

increases with the overexpression of SLC13A2 (Figs. 7F and 9C). The reason for this disparate result is that blood glucose has to be dynamically regulated catering to the requirement of liver regeneration. In the initiation of liver regeneration (1 d), decrease of blood glucose acts as an initial signal to trigger the start of liver regeneration, as reported previously that glucose supplementation suppresses hepatectomy-induced hepatocellular proliferation (Holecek, 1999; Huang and Rudnick, 2014; Weymann et al, 2009). After the initiation phase, metabolic adaptation occurs including suppression of liver glycolytic activity and shift of pyruvate metabolism, to obtain the maintenance of glucose homeostasis. In the propagation phase of liver regeneration, blood glucose acts as a predominant energy source for cell proliferation, which has huge energetic demand, as evidenced that prolonged hypoglycemia impaired liver regeneration and reduced mice survival after PHx (Alvarez-Guaita et al, 2020). Therefore, both decrease at 1 d and increase at 3 d of blood glucose induced by SLC13A2 overexpression should be beneficial for liver regeneration.

Citrate is also an important metabolite and metabolic regulator to inhibit tumor growth and progression by reprogramming lipid metabolism and determining cancer cell fate in various cancer types (Zhao et al, 2022). Also, suppressed citrate cleavage by the inhibition of ACLY leads to the decrease of acetyl-CoA and histone acetylation, resulting in the downregulation of proliferative and metastatic genes in cancer cells (Kumari et al, 2019). In another parallel study from our group, we also found that SLC13A2-transported citrate releases acetyl-CoA to suppress tumor growth through the acetylation and degradation of a key glycolytic enzyme pyruvate kinase muscle isozyme M2 (PKM2) (Qin et al, 2024). The disparate roles of citrate in cancer and liver regeneration indicate that pathologically uncontrolled cell proliferation has drastically different metabolic requirements with the energetic demands of general controlled organogenesis, even though they share similar properties in transcriptional level, especially the genes of DNA replication and cell division for proliferation (Fausto et al, 2006). It is interesting that even though liver regeneration shares similar transcriptomic signature with liver tumorigenesis, the metabolic reprogramming exhibits typical properties of these two liver pathologies respectively. How metabolism caters to the demand of cell proliferation in the two different contexts is an important question for providing insights into the liver pathophysiology and disease development. Even more interestingly, SLC13A2 exhibits equally beneficial effects in both scenarios by facilitating liver regeneration and ameliorating liver cancer progression. It suggests a novel model of killing two birds with one stone, which could be a potential target especially for the cancer patients after liver resection to remove tumor tissues.

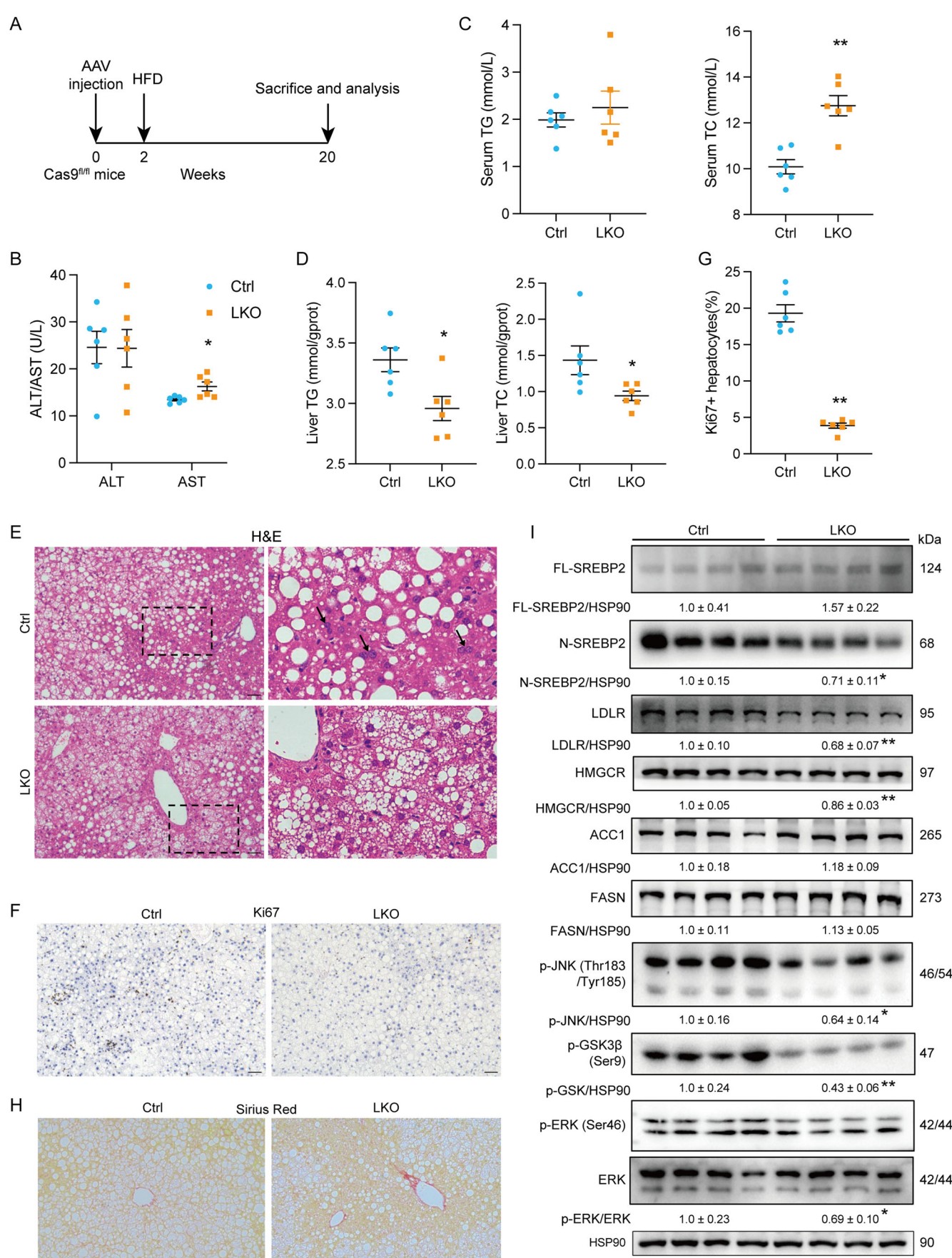

**Figure 6. Liver-specific knockout of SLC13A2 suppresses liver regeneration in chronic liver injury induced by HFD feeding.**

(A) Diagrams of liver-specific knockout (LKO) of SLC13A2 and study design. AAV-TBG-Cre-U6-Lacz (Ctrl) and AAV-TBG-Cre-U6-sgRNA (LKO) virus ($5 \times 10^{11}$ VP/mouse) were injected to the Rosa26-FSF-Cas9 knockin mouse via tail vein. The mice were subjected to HFD (60% kcal) for 20 weeks. (B) Serum ALT and AST levels ($N = 6$). (C, D) Serum concentrations (C) and liver contents (D) of TG/TC ($N = 6$). (E, F) Representative images of H&E staining (E) and Ki67 IHC staining (F). Arrow heads are pointing to the double-nucleated hepatocytes. Scale bar, 50 μm. (G) Quantification for the ratios of Ki67-positive cells ($N = 6$). (H) Representative images of sirius red staining. Scale bar, 50 μm. (I) Immunoblotting of proteins in signaling pathways and cholesterol synthesis ($N = 4$). Data are expressed as Mean ± SEM. $*P < 0.05$ and $**P < 0.01$ LKO vs Ctrl group, two-tailed unpaired Student's $t$ test. Source data are available online for this figure.

# Methods

### Reagents and tools table

| Reagent/Resource | Reference or Source | Identifier or Catalog Number |
|---|---|---|
| **Experimental models** | | |
| Mouse: Rosa26-flox-STOP-flox-Cas9 knockin mice | Jackson Lab | Cat#: 024857 |
| Mouse: male C57BL/6J mice | Center for Comparative Medicine, Yangzhou University, Yangzhou, China | N/A |
| **Recombinant DNA** | | |
| pAdeno-MCMV-SLC13A2-3Flag | This paper | N/A |
| PAd Delta F6 | Addgene | Cat#: 112867 |
| pAAV2/8 | Addgene | Cat#: 112864 |
| pX602 | Addgene | Cat#: 61593 |
| pX602-AAV-Cre-SgSLC13A2 | This paper | N/A |
| AAV-TBG-GFP | Addgene | Cat#: 105535 |
| AAV-TBG-SLC13A2 | This paper | N/A |
| **Antibodies** | | |
| Antibody | This paper | Appendix Table S2 |
| **Oligonucleotides and other sequence-based reagents** | | |
| gRNA sequence targeting mouse SLC13A2 coding exons | This paper | Appendix Table S1 |
| Sequences of primers for construction | This paper | Appendix Table S1 |
| Sequences of primers for qPCR analysis | This paper | Appendix Table S1 |
| **Chemicals, Enzymes and other reagents** | | |
| Materials and regents | This paper | Appendix Table S3 |
| **Software** | | |
| Image J | National Institutes of Health | https://imagej.nih.gov/ij/ |
| Graphpad prism 9 | Graphpad Software | https://graphpad.com |
| R software | R Foundation Statutes | https://www.r-project.org/ |

| Reagent/Resource | Reference or Source | Identifier or Catalog Number |
|---|---|---|
| **Cell line** | | |
| Mouse normal hepatocytes AML12 | Institute of Biochemistry and Cell Biology of the Chinese Academy of Sciences | N/A |
| HEK293T | ATCC | Cat#: CRL-11268; RRID: CVCL_1926 |

Reagents including cell lines, antibodies, and plasmid DNAs used in this study are listed in Reagents and Tools table and Appendix Tables S1, S2 and S3.

## Animal studies

All animal protocols were approved by the IACUC (Institutional Animal Care and Use Committee) of China Pharmaceutical University (Approval number: 2021-07-016), which are in accordance with the National Institutes of Health guidelines. In the experiments with liver-specific knockout, *Rosa26-flox-STOP-flox-Cas9* knockin mice (*Rosa26-LSL-Cas9*, Jackson, #024857) were used and single-guide RNA (sgRNA) targeting SLC13A2 (Appendix Table S1) were constructed under the control of U6 promoter, along with thyroid hormone-binding globulin (TBG)-Cre, into adeno-associated virus 8 (AAV8, $5 \times 10^{11}$ genome copies/mouse) as previously described (Chu et al, 2016). In the experiments with liver-specific overexpression, male wild-type C57BL/6J mice from University of Yangzhou aged 6–8 weeks were randomly grouped and administrated AAV8 (GFP vs SLC13A2, $1 \times 10^{11}$ genome copies/mouse) with liver-specific TBG promoter through tail vein injection. The primers used for construction were shown in Table S1. For chronic mild liver injury, mice were fed high-fat diet (HFD) containing 60% kcal% fat (Research Diet, D12492) for 20 weeks. All procedures were carried out to minimize the sufferings of the animals.

## Surgery of 70% partial hepatectomy

A surgery of 70% PHx was performed with modification as described previously (Hu et al, 2020; Mitchell and Willenbring, 2008). Briefly, mice were anesthetized with 2% isoflurane, and the median and left lateral hepatic lobes were removed with ligation before resection. Remnant livers were harvested for analysis at indicated time points after the surgery. Mice in sham group were performed with surgery of opening abdominal cavity without liver lobe resection. The efficiency of liver regeneration was determined

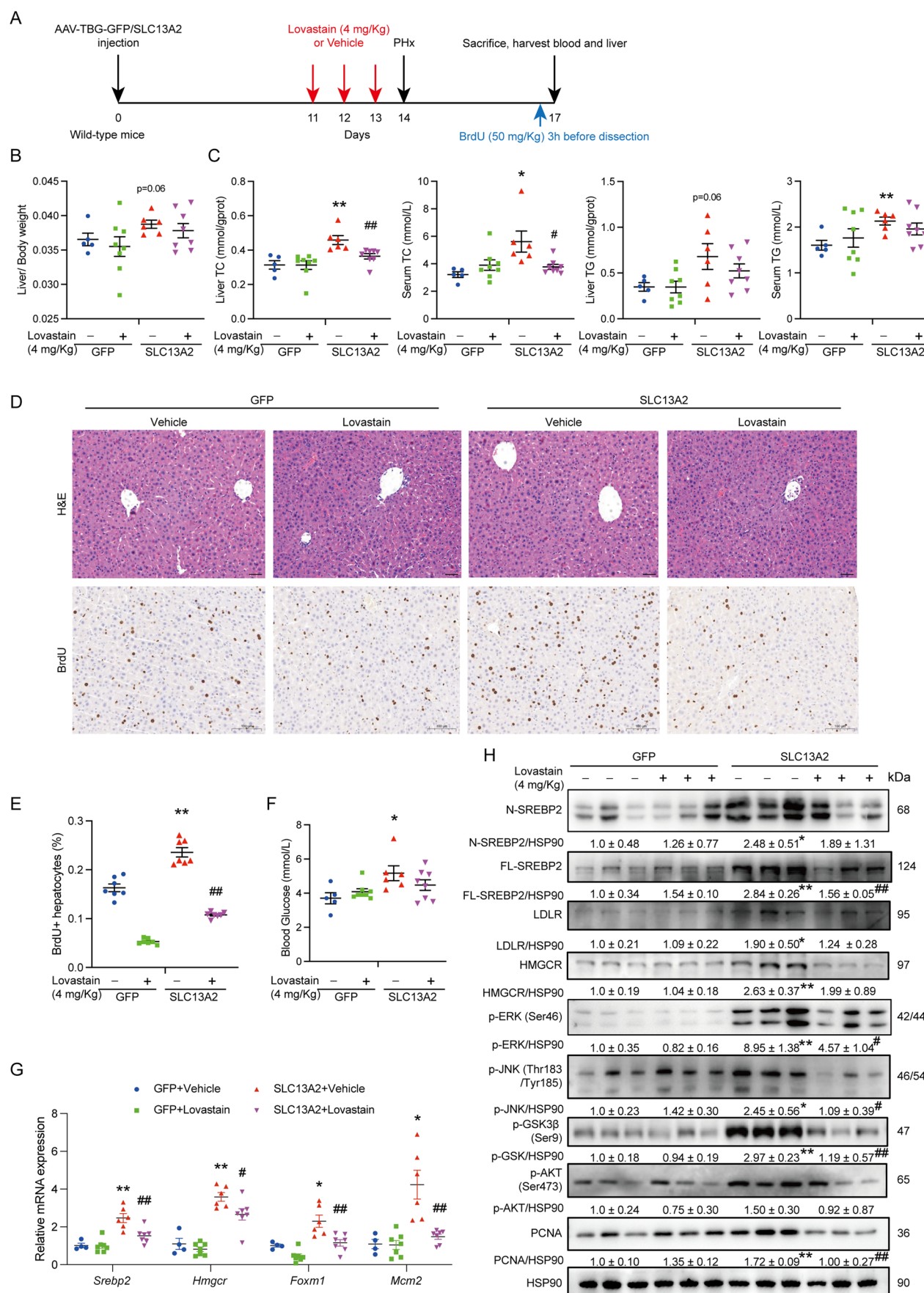

**Figure 7. SLC13A2 promotes liver regeneration through the activation of cholesterol synthesis.**

(A) Diagrams of study design. Mice were injected with AAV virus (GFP: AAV-TBG-GFP; SLC13A2: AAV-TBG-SLC13A2 virus ($1 \times 10^{11}$ VP/mouse) through tail vein to ensure the liver-specific overexpression of SLC13A2, followed by the intragastrical administration of Lovastain (4 mg/Kg) or vehicle once a day for consecutive 3 days. Then the surgery of PHx was operated, and the mice were sacrificed after 3 days. (B) Liver weight versus body weight ratio. (C) Liver contents and serum concentrations of TG/TC. (D) Representative images of H&E staining (scale bar, 50 μm) and BrdU incorporation (scale bar, 100 μm). (E) Ratio of BrdU-positive hepatocytes ($N = 7$). (F) Blood glucose. (G) RNA expression of genes involving cholesterol synthesis and cell proliferation. (H) Immunoblotting of proteins in signaling pathways and cholesterol synthesis. Data are expressed as Mean ± SEM (GFP, $N = 5$; SLC13A2, $N = 6$; GFP+Lovastatin or SLC+Lovastatin, $N = 8$). *$P < 0.05$ and **$P < 0.01$ SLC13A2 vs GFP group, #$P < 0.05$ and ##$P < 0.01$ Lovastatin vs Vehicle group, two-way ANOVA followed by Bonferroni multiple comparisons. Source data are available online for this figure.

by liver/body weight ratio. For BrdU labeling, mice were administrated intraperitoneally BrdU (50 mg/kg) 3 h before sacrifice. The dosing regimen and timing for the animals were detailed in the figure legend.

## Serum and liver biochemical analysis

Blood was collected before the dissection and serum were prepared with centrifugation at $5000 \times g$ at 4 °C for 10 min. Approximately 100 mg of liver tissue was grounded with a 1:9 weight/volume ratio of methanol and centrifuged at $2500 \times g$ at 4 °C for 10 min. Triglycerides (TG) and total cholesterol (TC) in liver homogenates and serum, alanine aminotransferase (ALT) and aspartate aminotransferase (AST) in serum were measured using assay kits from Jiancheng (Nanjing, China). Protein concentrations were quantified using a BCA protein assay kit (Vazyme, China) and used for the normalization of liver lipid contents.

## Cell culture

Mouse primary hepatocytes were isolated from wild-type male C57BL/6J mice, SLC13A2 control (Ctrl) or liver-specific SLC13A2 knockout (LKO) mice and cultured as described previously (Xiong et al, 2020). Mouse normal hepatocyte cell line AML12 was obtained from the Institute of Biochemistry and Cell Biology of the Chinese Academy of Sciences (Shanghai, China), characterized by short tandem repeat profiling and cultured as previously described (Lv et al, 2022). Adenoviral infection (pAdeno-MCMV-3XFlag versus pAdeno-MCMV-SLC13A2-3XFlag, OBiO, Shanghai, China) was performed on the same day of primary hepatocytes isolation or the next day of AML12 cells plating. Hepatocytes were switched to fresh Duldecco's modified Eagle's medium (DMEM) with 10% FBS after 12 h and then allowed transduction for designated period. Primary hepatocytes isolated from Ctrl or LKO mice were cultured overnight to allow adherence and used for following experiments. Citrate (200 μM) or vehicle was applied to primary hepatocytes in DMEM medium with 2% lipoprotein deficient serum (LPDS) for 24 h to stimulate cholesterol synthesis. All cells were negative of mycoplasma contamination.

## Isolation of subcellular fractions

Primary hepatocytes from wild-type mice were transduced with SLC13A2-overexpression adenovirus for 60 h, followed by the isolation procedure with modification from previously described protocol (Clayton and Shadel, 2014). Briefly, cell pellets were resuspended in 1 ml of ice-cold homogenization buffer (225 mM mannitol, 75 mM sucrose, 0.1 mM EDTA, 30 mM Tris-HCl,

pH 7.4) and a portion was taken to prepare whole-cell lysate. The remaining suspension was transferred to a 2-mL Dounce homogenizer (pre-cooled to 4 °C) and grinded for 5 to 10 times, ensuring the proportion of whole cell is ≤10% under a microscope. Centrifuge the homogenate at $600 \times g$ at 4 °C for 5 min to pellet the nuclei, which was resuspended in 1 ml of ice-cold homogenization buffer and washed twice to obtain the nuclei. Transfer the supernatant from the last step to a new tube and centrifuge at $7000 \times g$ for 10 min at 4 °C to pellet the mitochondria. Resuspend the mitochondria in ice-cold homogenization buffer without EDTA (225 mM mannitol, 75 mM sucrose, 30 mM Tris-HCl, pH 7.4) and wash twice to obtain the mitochondria. Transfer the supernatant from the previous step to a new tube, sit on ice for 1.5 h, and then centrifuge at $20,000 \times g$ for 30 min at 4 °C to pellet the cytoplasmic membrane, leaving the supernatant as cytosol.

## Quantitative reverse transcriptase–polymerase chain reaction (qRT–PCR) and immunoblotting

Total RNA was extracted using TRIzol reagent (Vazyme, Nanjing, China). cDNA was then synthesized using the PrimeScript™ RT Reagent Kit (Vazyme, Nanjing, China). qRT-PCR was performed with the qPCR SYBR Green Master Mix (High ROX Premixed) kit (AG Bio, Hunan, China) on a fluorescence quantitative PCR machine (Applied Biosystems, USA). The primers designed with NCBI primer BLAST are shown in Table S1. Protein samples were quantified, separated, and transferred onto a polyvinylidene fluoride membrane. The membranes were incubated with primary antibodies (Appendix Table S2) at 4 °C overnight and then with secondary antibodies for 1 h at room temperature. Enhanced chemiluminescence reagents were used to visualize the target proteins.

## Histological and immunohistological chemistry staining

Hematoxylin and eosin were stained on formalin-fixed and paraffin-embedded liver sections (H&E staining). Immunohistochemical staining for BrdU and Ki67 was performed on paraffin-embedded liver tissue sections using a DAB Horseradish Peroxidase Color Development kit (Beyotime, Shanghai, China) according to the manufacturer's instructions. Images were taken and digitalized with an Olympus microscope equipped with a BX53 digital camera (Olympus, Japan).

## Cell proliferation assays and cholesterol staining in vitro

For the 5-ethynyl-2'-deoxyuridine (EdU) incorporation assay, AML12 cells were plated on a 24-cell plate at a density of $2 \times 10^3$ cells/well,

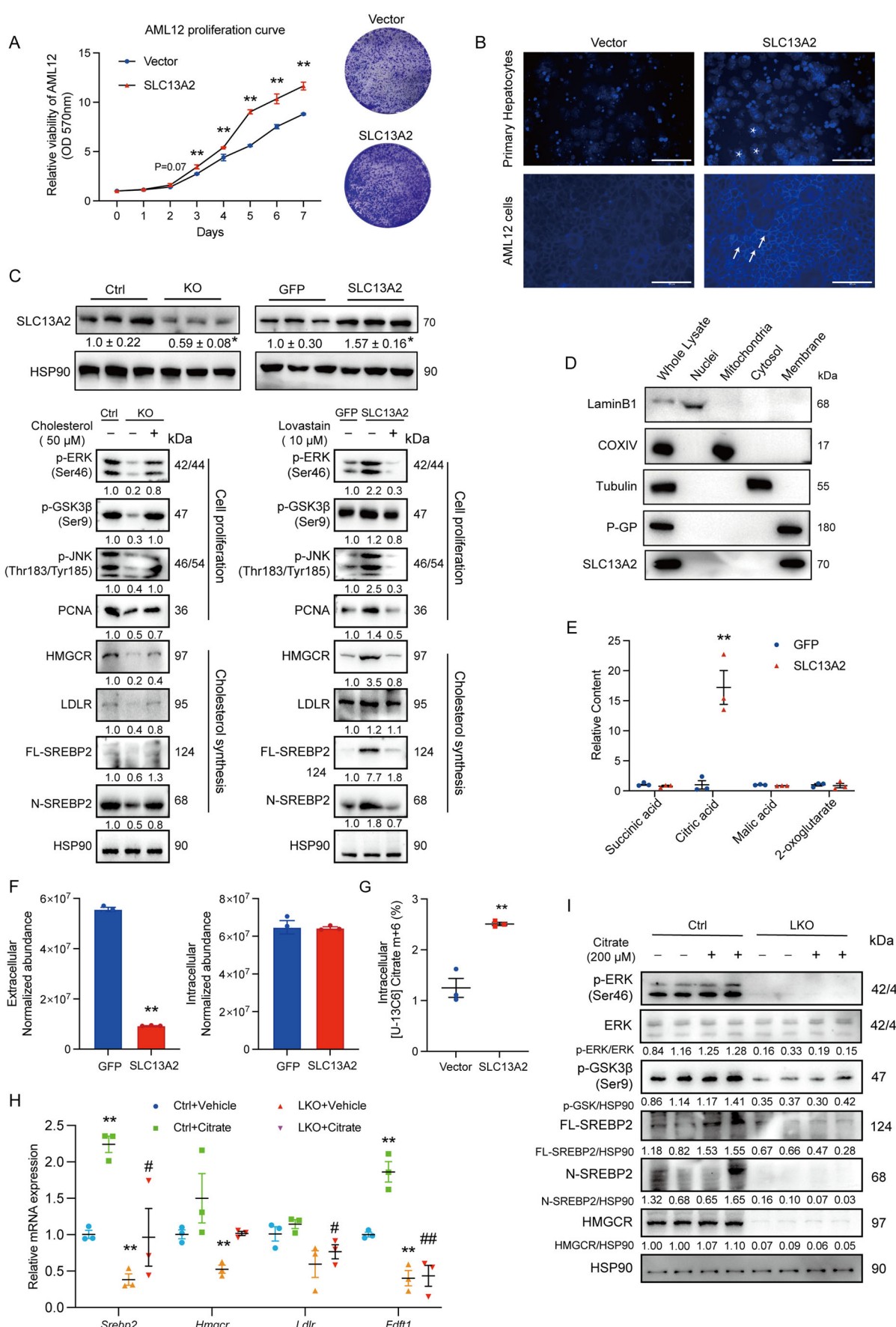

**Figure 8.  SLC13A2 promotes cell proliferation through the import of citrate for cholesterol synthesis as a membrane transporter in hepatocytes.**

(A) Cell viability (left) and colony formation capability (right) of AML12 cells 48 h after the infection with adenovirus, and cultured for designated days (cell viability) or 7 days (colony formation) (N = 3). (B) AML12 cells and primary hepatocytes were transduced with adenovirus for 48 h, and then stained with fluorescent dyes for cholesterol staining. Stars or arrow heads indicated cholesterol staining in hepatocytes or AML cells, respectively. Scale bar, 50 μm. (C) Cas9$^{fl/fl}$ mice were injected with AAV virus (AAV-TBG-Cre-U6-LACZ, Ctrl; AAV-TBG-Cre-U6-SLC13A2 gRNA, LKO) through tail vein to ensure the liver-specific knockout of SLC13A2 in LKO group; primary hepatocytes were isolated, allowed adherence overnight, and cultured in the presence of vehicle or cholesterol (50 μM) for 48 h (left). Primary hepatocytes were isolated from wild-type mice and then transduced with adenovirus (vector vs SLC13A2) for 48 h, followed by the treatment with vehicle or Lovastatin (10 μM) for 48 h (right). (D) Cellular fractions were isolated from primary hepatocytes with SLC13A2-overexpressing adenovirus transduction for 60 h, followed by immunoblotting for the cellular localization of SLC13A2 with specific controls. (E) Putative substrates for SLC13A2 transportation in the liver tissues from mice transduced with AAV at 1 d after PHx surgery were measured using UPLC-MS analysis (N = 3). (F) Primary hepatocytes were cultured and transduced with adenovirus for 60 h, then the abundance of citrate in both extracellular and intracellular compartment were analyzed using UPLC-MS analysis (N = 3). (G) Primary hepatocytes with overexpression of SLC13A2 were treated with 125 μM [U-13C6] citrate for 1 h. The intracellular metabolites were then extracted for targeted metabolomics analysis of the m + 6 citrate ratio (N = 3). (H, I) Primary hepatocytes were isolated from Ctrl or LKO mice. After overnight culture for adherence, the cells were treated with 200 μM citrate for 24 h in DMEM medium with 2% LPDS. RT-qPCR and immunoblotting assays were utilized to analyze the expression of cholesterol synthesis genes (H, N = 3) as well as the activation of cellular signaling pathways and cholesterol synthesis (I). Data are expressed as Mean ± SEM. *P < 0.05 and **P < 0.01 SLC13A2 vs Vector/GFP, LKO vs Ctrl, citrate vs vehicle; $^{#}$P < 0.05 and $^{##}$P < 0.01 LKO+citrate vs Ctrl+citrate; two-tailed unpaired Student's t test (E–G), one-way (H) or two-way (A) ANOVA followed by Bonferroni multiple comparisons. Source data are available online for this figure.

incubated with EdU (10 μM) for 2 h, fixed for 15 min and analyzed using a commercial kit (Beyotime, Shanghai, China). For cell viability assay, AML12 cells were split on a 96-cell plate at a density of $1 \times 10^3$ cells/well and cultured as designated days and analyzed using a CCK8 assay kit (Vazyme, Nanjing, China) according to the manufacturer's instructions. For colony formation assay, AML12 cells were split onto a 6-well plate at a density of $1 \times 10^3$ cells/well, and cultured for 7 days. For cholesterol staining, cells were rinsed with PBS and fixed with fresh paraformaldehyde for 1 h, followed by the staining with filipin working solution (0.05 mg/mL) for 30 min at room temperature. The images were quickly captured using fluorescence microscopy with a UV filter set (Meshot, China).

## Metabolomic analysis

Sample extraction and detection methods for both targeted and untargeted metabolomics analyses were consistent. For liver tissue samples, about 30 mg of liver tissues were quenched in liquid nitrogen, added with 800 μl of ice-cold methanol containing 1 μg/mL 4-chloro-DL-phenylalanine, homogenized, shaken for 10 min, and put in a −80 °C freezer for 2 h to ensure complete lysis, then centrifuged at $18,000 \times g$ for 10 min at 4 °C. The supernatant was evaporated to dry using a vacuum centrifugal concentrator (Thermo, Massachusetts, USA). Normalized metabolite abundance values were further normalized to the actual liver weights for subsequent quantification. Hepatocytes from wild-type mice were transduced with adenovirus for 60 h. For targeted metabolomics analysis, the SLC13A2-overexpressed hepatocytes were treated with 125 μM [U-13C6] citric acid for 1 h. Cell metabolites were also dissolved in ice-cold 80% methanol containing 1 μg/mL 4-chloro-DL-phenylalanine. The samples were then incubated for 30–40 min in a −80 °C freezer, followed by centrifugation at $15,000 \times g$ for 20 min at 4 °C, after which the supernatant was dried. Before analysis, the samples were reconstituted with methanol, followed by centrifugation at $18,000 \times g$ for 10 min at 4 °C twice, and conducted using a 6546 LC/Q-TOF (Agilent, USA) instrument equipped with an electrospray ionization (ESI) source. Separation was achieved on an XBridge Premier BEH Amide VanGuard FIT column (2.5 μm, 4.6 mm × 150 mm, Waters, USA). MS data were acquired in negative ESI mode over a range of m/z 50 to 850. The mobile phase consisting of water (containing 0.3% ammonium acetate and 0.3%

ammonia in phase A, v/v) and acetonitrile (phase B) was delivered at a flow rate of 0.4 ml min$^{-1}$. Gradient elution was conducted as follows: 85% B from 0–1 min, 85–30% B from 1–12 min, 30% B from 12–14 min, 30–85% B from 14–14.5 min, 85% B from 14.5–36 min. Mass Hunter Workstation (ver. B.06.00, Agilent Technologies) software was used for targeted metabolomics data acquisition and analysis. The untargeted metabolomics data were analyzed by Progenesis QI.

For lipid metabolites extraction, cells were split and collected, followed by the addition of 1 mL methyl tert-butyl ether and shake at 37 °C 1400 rpm for 15 min. Then, 250 μL ultra-pure water was added and kept balance for 10 min at 4 °C. Organic layer supernatant was taken after twice centrifugation 14,000 rpm for 20 min at 4 °C and then dried by nitrogen blowing instrument (Anpel, Shanghai, China) for subsequent liquid phase mass spectrometry analysis. Lipidomics analysis was conducted using a SYNAPT G2-Si HDMS Mass spectrometry analysis system (Waters, USA) instrument equipped with an ESI source. Separation was achieved on an Acquity UPLC BEH C18 column (1.7 μm, 50 × 2.1 mm, Waters, USA). MS data were acquired in positive ESI mode over a range of m/z 50 to 1200. The mobile phase consisting of water (containing 0.3% formic acid in phase A, v/v) and methanol (phase B) was delivered at a flow rate of 0.3 ml min$^{-1}$. Gradient elution was conducted as follows: 5–10% B from 0–1 min, 10–45% B from 1–18 min, 45–90% B from 18–20 min, 90% B from 20–21 min, 90–45% B from 21–21.5 min, 45–10% B from 21.5–23 min, 10–5% B from 23–30 min. The metabolomics data were analyzed by Progenesis QI with conditions of P < 0.05, fold change > 2, and VIP > 1 to screen for differentially abundant metabolites. Metabolic pathway analysis was performed based on the MetaboAnalyst 5.0 platform.

## Bioinformatic analysis

The published RNA-Seq raw data of mouse liver samples at time points in the initiation and progression phases of liver regeneration after partial hepatectomy were downloaded from the Gene Expression Omnibus database (GSE95135 20 h, GSE215423 24 h, GSE211370 72 h, GSE188421 24–72 h, GSE188768 24 h, GSE204901 48 h). The datasets with read count were analyzed for differential gene expression using the DESeq2 package (Love et al,

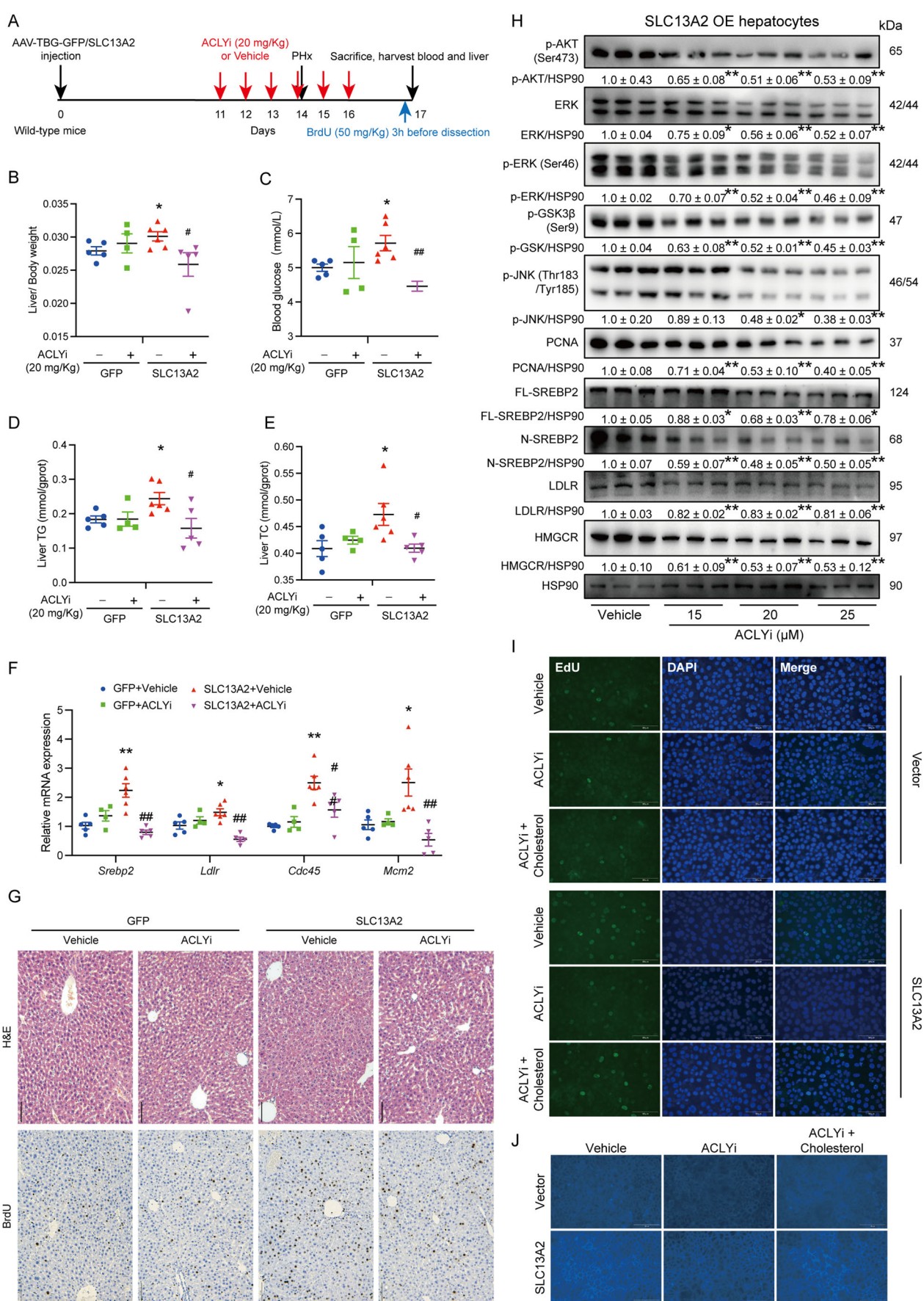

**Figure 9.   Citrate transported by SLC13A2 is catalyzed by ACLY, which secretes Acetyl-CoA for cholesterol synthesis to facilitate liver regeneration.**

(A) Diagrams of study design. Mice were injected with AAV virus (GFP: AAV-TBG-GFP; SLC13A2: AAV-TBG-SLC13A2 virus; $1 \times 10^{11}$ VP/mouse) through tail vein to ensure the liver-specific overexpression of SLC13A2. ACLYi (20 mg/Kg) or vehicle were administrated intragastrically once daily for consecutive 3 days. The surgery of PHx was performed on the third day and followed by 3 more doses of ACLYi administration. The mice were sacrificed for analysis at 3 d after the surgery. (B) Liver weight versus body weight ratio. (C) Blood glucose. (D, E) Liver TG (D) and TC (E) content. (F) RNA expression of genes involving cholesterol synthesis and cell proliferation. (G) Representative images of H&E staining and BrdU incorporation. Scale bar, 100 μm. (H) Primary hepatocytes were cultured and transduced with adenovirus for 48 h, followed by the the treatment with ACLYi in different concentrations (15, 20 and 25 μM) for 48 h ($N = 3$). Scale bar, 50 μm. (I) AML12 cells were transduced with adenovirus for 48 h, followed by the treatment of ACLYi (25 μM) in the presence or absence of cholesterol (50 μM) for 48 h. EdU incorporation assay was performed with the addition of EdU for 2 h. Scale bar, 50 μm. (J) Cholesterol staining using filipin fluorescence dye were performed in AML12 cells after adenovirus transduction for 48 h followed by the treatment of ACLYi (25 μM), in the absence or presence of cholesterol (50 μM) for 48 h. Scale bar, 50 μm. Data are expressed as Mean ± SEM (GFP, $N = 5$; GFP+ACLYi, $N = 4$; SLC13A2, $N = 6$; SLC+ACLYi, $N = 5$). *$P < 0.05$ and **$P < 0.01$ SLC13A2 vs GFP, ACLYi vs vehicle (H); #$P < 0.05$ and ##$P < 0.01$ ACLYi vs vehicle (B–F); one-way ANOVA followed by Bonferroni multiple comparisons. Source data are available online for this figure.

2014), whereas the ones with FPKM or RPKM were analyzed using the limma package in the R program (Ritchie et al, 2015). Mouse gene annotation packages were used to convert ENSEMBL IDs to gene symbols, and genes with a log2FoldChange > 1 or <1 and a *P*-value less than 0.05 were defined as differentially expressed genes. Heatmaps and volcano plots were generated using the pheatmap and ggplot2 packages by R program.

## Statistical analysis

The data are expressed as mean ± SEM. Statistical significance was considered using unpaired two-tailed Student's *t* test between two groups and one-way ANOVA followed by Bonferroni multiple comparisons among groups more than two. Differences from cell viability study were analyzed by two-way ANOVA, followed by Bonferroni multiple comparisons. $P < 0.05$ indicates statistical significance.

## Data availability

All data are available in the source data and appendix tables. This study includes no data deposited in external repositories.

The source data of this paper are collected in the following database record: biostudies:S-SCDT-10_1038-S44318-025-00362-y.

## Peer review information

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

## Acknowledgements

The study was supported by the National Natural Science Foundation of China (No. 82273982 and 82070883 to JX), the Natural Science Foundation of Jiangsu province (No. BK20221525 to JX) and Scientific Research Foundation for high-level faculty, China Pharmaceutical University (to JX).

## Author contributions

**Li Shi**: Investigation; Methodology. **Hao Chen**: Formal analysis; Investigation; Visualization; Methodology; Writing—original draft. **Yuxin Zhang**: Investigation; Methodology. **Donghao An**: Data curation; Software; Investigation; Visualization. **Mengyao Qin**: Investigation. **Wanting Yu**: Investigation; Visualization. **Bin Wen**: Investigation. **Dandan He**: Methodology. **Haiping Hao**: Resources; Supervision; Writing—review and editing. **Jing Xiong**: Conceptualization; Resources; Data curation; Software; Formal analysis; Supervision; Funding acquisition; Validation; Investigation; Visualization; Methodology; Writing—original draft; Project administration; Writing—review and editing.

Source data underlying figure panels in this paper may have individual authorship assigned. Where available, figure panel/source data authorship is listed in the following database record: biostudies:S-SCDT-10_1038-S44318-025-00362-y.

## Disclosure and competing interests statement

The authors declare no competing interests.

