## [Peer Review File · The EMBO Journal]

SLC13A2 promotes hepatocyte metabolic remodeling and liver regeneration by enhancing de novo cholesterol biosynthesis

Li Shia, Hao Chena, Yuxin Zhanga, Donghao Ana, Mengyao Qina, Wanting Yua, Bin Wena, Dandan Heb, Haiping Haoa, Jing Xionga

Corresponding authors: Jing Xiong (jxiong@cpu.edu.cn) , Jing Xiong (jxiong@cpu.edu.cn)

Review Timeline:

Submission Date:	9th Jul 24
Editorial Decision:	17th Aug 24
Revision Received:	2nd Nov 24
Editorial Decision:	5th Dec 24
Revision Received:	6th Dec 24
Accepted:	11th Dec 24

Editor: Daniel Klimmeck

Transaction Report:

Dear Dr Xiong,

Thank you for the submission of your manuscript (EMBOJ-2024-118432) to The EMBO Journal as well for your patience with our response at this time of the year. The delay was due to protracted referee input and detailed discussion in the editorial team. Your study was assessed by two reviewers with expertise in liver biology and lipid metabolism, whose comments are enclosed below.

As you will see from the experts' reports, the referees acknowledge the analysis and potential interest of your results. However, they also express major concerns regarding completeness and detail of the findings, which need to be addressed thoroughly to make them supportive of publication in the EMBO Journal. The reviewers also raise a number of issues related to the data presentation, additional controls and improved methods annotation required, statistics applied and overall discussion of related literature, that would need to be conclusively addressed to achieve the level of robustness and clarity needed for The EMBO Journal.

Given the overall interest stated and broader angle of your findings, we are able to invite you to revise your manuscript experimentally to address the referees' comments. I need to stress though that we do require strong support from the referees on a revised version of the study in order to move on to publication of the work. Given the open outcome of the revision experiments, I suggest keeping EMBO Reports in mind as an alternative venue for this work.

In light of the extensive experimentation requested, I would appreciate if you could contact me during the next weeks for exchange e.g. a video call to discuss your perspective on the comments and potential plan for revisions.

Please feel free to contact me if you have any questions or need further input on the referee comments.

When submitting your revised manuscript, please carefully review the instructions below.

Please feel free to approach me any time should you have additional questions related to this.

Thank you for the opportunity to consider your work for publication.

I look forward to your revision.

Kind regards,

Daniel Klimmeck

Daniel Klimmeck, PhD
Senior Editor
The EMBO Journal

Instruction for the preparation of your revised manuscript:

- 1) a .docx formatted version of the manuscript text (including legends for main figures, EV figures and tables). Please make sure that the changes are highlighted to be clearly visible.
- 2) individual production quality figure files as .eps, .tif, .jpg (one file per figure).
- 3) a .docx formatted letter INCLUDING the reviewers' reports and your detailed point-by-point response to their comments. As part of the EMBO Press transparent editorial process, the point-by-point response is part of the Review Process File (RPF), which will be published alongside your paper.
- 4) a complete author checklist, which you can download from our author guidelines (<https://wol-prod-cdn.literatumonline.com/pb->

assets/embo-site/Author Checklist%20-%20EMBO%20J-1561436015657.xlsx). Please insert information in the checklist that is also reflected in the manuscript. The completed author checklist will also be part of the RPF.

6) It is mandatory to include a 'Data Availability' section after the Materials and Methods. Before submitting your revision, primary datasets produced in this study need to be deposited in an appropriate public database, and the accession numbers and database listed under 'Data Availability'. Please remember to provide a reviewer password if the datasets are not yet public (see <https://www.embopress.org/page/journal/14602075/authorguide#datadeposition>).

7) Our journal encourages inclusion of *data citations in the reference list* to directly cite datasets that were re-used and obtained from public databases. Data citations in the article text are distinct from normal bibliographical citations and should directly link to the database records from which the data can be accessed. In the main text, data citations are formatted as follows: "Data ref: Smith et al, 2001" or "Data ref: NCBI Sequence Read Archive PRJNA342805, 2017". In the Reference list, data citations must be labeled with "[DATASET]". A data reference must provide the database name, accession number/identifiers and a resolvable link to the landing page from which the data can be accessed at the end of the reference. Further instructions are available at .

8) At EMBO Press we ask authors to provide source data for the main and EV figures. Our source data coordinator will contact you to discuss which figure panels we would need source data for and will also provide you with helpful tips on how to upload and organize the files.

Numerical data can be provided as individual .xls or .csv files (including a tab describing the data). For 'blots' or microscopy, uncropped images should be submitted (using a zip archive or a single pdf per main figure if multiple images need to be supplied for one panel). Additional information on source data and instruction on how to label the files are available at .

9) We replaced Supplementary Information with Expanded View (EV) Figures and Tables that are collapsible/expandable online (see examples in <https://www.embopress.org/doi/10.15252/emj.201695874>). A maximum of 5 EV Figures can be typeset. EV Figures should be cited as 'Figure EV1, Figure EV2' etc. in the text and their respective legends should be included in the main text after the legends of regular figures.

11) For data quantification: please specify the name of the statistical test used to generate error bars and P values, the number (n) of independent experiments (specify technical or biological replicates) underlying each data point and the test used to calculate p-values in each figure legend. The figure legends should contain a basic description of n, P and the test applied. Graphs must include a description of the bars and the error bars (s.d., s.e.m.).

The revision must be submitted online within 90 days; please click on the link below to submit the revision online before 15th Nov 2024.

Referee #1:

Dr. Xiong reported a comprehensive study on the identification of SLC13A2 in PHx induced liver regeneration via regulating the cholesterol pathway. The story is interesting and covered many aspects in terms of the transportation of citrate for cholesterol synthesis, liver regeneration, polyploid, and metabolic transcription regulation. However, the rather diffuse study cannot cover all aspects in depth and the missing of biological significance of SLC13A2 in overall liver growth and mice survival upon injury or diseases.

I have the following concerns:

1. Does SLC13A2 have the general role in liver regeneration, or it is PHx specific?
2. What is the consequence of loss of SLC13A2 upon liver injury for a long term in terms of liver function and hepatocyte proliferation? Does the SLC13A2 deletion delay the regenerative process only for a short time followed by compensation from other pathways? Authors should have included data of a longer time point to show the liver function, liver mass, lipid metabolism, survival if there is a difference.
3. Fig3. What is the phenotype of SLC13A2 overexpression during homeostasis before PHx?
4. Fig5. How does SLC13a2 regulate the cholesterol synthesis genes, the involve of mRNA expression level or the SLC13A2 transporter on cell membrane? For the overexpression study, authors should show the protein distribution in cells.
5. Figure legend and method sections need more details about the experiments, cell types and treatment. Fig7b staining need to be marked clearly in the figure, and Fig7a standard deviation is missing.
6. Fig6H and E, G, lovastatin treatment showed increased cell proliferation, p-JNK pathway, and FL-Srebp2 in H but not in E and G.
7. Fig6. how does the data look like in 7 days after PHx? 3 days has no obvious difference of LW/BW ratio as shown in 7 days
8. Fig7C,D, treatment of cholesterol upon KO should be blot together with the control group. Like this, authors can evaluate the contribution of the cholesterol treatment compared to the control group. Specifically, the Upper and bottom panels should be blot together to check the contribution of cholesterol in proliferation and signaling pathways listed.
9. How is the in vivo phenotype of ACLYi? This is to evaluate if citrate transportation is a key mediator of SLC13A2 mediated liver regeneration.
10. Along the same line as 9, Authors should check the contribution of Citrate in vitro and in vivo in terms of liver regeneration. Knockout SLC13A2 and treat with citrate in hepatocytes and check if SLC13A2 is the major transporter for citrate during liver regeneration.

Referee #2:

Hepatocytes undergo metabolism remodeling during liver regeneration, in this manuscript Shi and colleges illustrated significance of cholesterol metabolism mediated by SLC13A2 for liver regeneration. The authors demonstrated SLC13A2 promote liver regeneration after partial hepatectomy (PHx) with knockout and overexpression mouse model induced by Adeno-Associated Virus (AAV) transfection. Mechanically, SLC13A2 encode multiple transmembrane transporters, which transport citrate into intracellular for de novo cholesterol synthesis and finally promote liver regeneration.

Major points:

- 1)In Figure 1A, the authors analyzed transcriptomic profiles from other papers and SLC13A2 was choose for the candidate genes from differently expressed genes during liver regeneration. But different time points are on behalf of different stage of liver regeneration, the author should introduce the screen strategy in detail, it is best to provide gene list of every part of Venn diagram.
- 2)The authors should explain the function of EGF treatment for primary hepatocyte in Figure 5C.
- 3)There is a feedback regulation in cholesterol metabolism. Enough cholesterol inhibits SREBP2 hydrolysis, then reduce nSREBP2 level and transcript of Hmgcr. But in Figure 7C (bottom left), why cholesterol supplement result in the rise of HMGCR and nSREBP2?
- 4)SLC13A2 play an essential role in initiation of liver regeneration, the authors should discuss that why the SLC13A2 down-regulated after PHx in Figure 1.
- 5)In Figure 3E, overexpression of SLC13A2 reduce the blood glucose level after PHx, but in Figure 6F, SLC13A2 increase the

blood glucose. Meanwhile, result description in manuscript of Figure 3E (Liver/body weight ratio was significantly increased, while blood glucose at 1 day after the surgery was decreased with the overexpression of SLC13A2, indicating accelerated restoration of liver mass and functions by SLC13A2) and 6F (The increase of blood glucose reflecting the restoration of liver function was also induced by SLC13A2, which was suppressed by the pretreatment of lovastatin) is also opposite.

Minor Points:

- 1)The authors should provide the specific time point of different transcriptome profile in Figure 1B.
- 2)In Figure 1G, it seemed no significant change in PCNA after PHx, especially in PHx3d which is the proliferative stage of liver regeneration. The author should check it again.
- 3)It is recommended that changing HSP90 (Top left) to ACTIN in Figure 4B to keep the consistency of reference gene in one figure.
- 4)The authors showed metabolomic data in primary hepatocytes after SLC13A2 overexpression, but no further exploration in the following works. The authors should explain more reason of this metabolomic data in manuscript.

Response to the Reviewers' Comments

Ms. Ref. No: EMBOJ-2024-118432 SLC13A2 connects transcriptomic regulation and metabolic remodeling for liver regeneration in mice

Dear Editor,

We are truly grateful to yours and the reviewers' valuable and thought-provoking comments and suggestions. Those comments and suggestions are valuable and helpful for revising and improving our manuscript. We have studied comments carefully and made corresponding modifications which we hope meet with approval. Revised portions are marked in the manuscript. Please note:

1. All changes (point by point) in the revised manuscript can be found according to the following Response to the Reviewers' Comments.
2. Black words are comments and suggestions from Reviewers and blue words are response to the comments and suggestions.
3. Because of the modifications of manuscript, the order of the figure and the references has been updated in the revised manuscript.

Comments from Referee #1:

Dr. Xiong reported a comprehensive study on the identification of SLC13A2 in PHx induced liver regeneration via regulating the cholesterol pathway. The story is interesting and covered many aspects in terms of the transportation of citrate for cholesterol synthesis, liver regeneration, polyploid, and metabolic transcription regulation. However, the rather diffuse study cannot cover all aspects in depth and the missing of biological significance of SLC13A2 in overall liver growth and mice survival upon injury or diseases.

We appreciate the reviewer's positive comments on our manuscript.

1. Does SLC13A2 have the general role in liver regeneration, or it is PHx specific?

The reviewer made an excellent point. In our study, we conclude that the role of SLC13A2 on liver regeneration is not PHx-specific. Besides PHx-induced liver regeneration, we also explored the effects of SLC13A2 on liver regeneration in a different mouse model with chronic mild liver injury induced by long-term feeding of high-fat diet (HFD) in the mice of liver-specific SLC13A2 deficiency.

There was no significant difference in body weight, blood glucose, liver weight, adipose tissue weight, glucose and insulin tolerance between the two groups. Interestingly, serum AST activity was increased significantly in LKO mice, indicating deteriorated liver injury (Figure 6B). In mice with SLC13A2 disruption, serum TC raised (Figure 6C), while liver TG and TC content declined (Figure 6D), in accordance with the suppressed expression of LDLR (Figure 6I), a receptor for serum cholesterol transportation to the liver. Akin to the effects of SLC13A2 on liver regeneration induced by PHx, the number of hepatocytes with double nuclei shrank (Figure 6E), and the percentage of cells with nuclear Ki67 positivity also reduced (Figure 6F-G) when liver specifically deleted with SLC13A2. More intriguingly, sirius red staining revealed increased collagen fibers in the mouse liver with SLC13A2 deficiency (Figure 6H), indicating suppressed regenerative capacity resulted in hepatocyte injury and hepatic fibrosis, especially under chronic damaging conditions (Haideri et al. 2017). Also, the maturation of SREBP2, expression of LDLR, as well as the signaling for hepatocellular proliferation, phosphorylation of JNK, ERK and GSK3 β , were all suppressed by SLC13A2 deficiency (Figure 6I). The data prove that SLC13A2 also enhances liver regeneration in HFD-induced chronic liver injury by multiplying cholesterol synthesis for hepatocellular proliferation.

The new Figure 6 is shown as follows.

Figure 6. Liver-specific knockout of SLC13A2 suppresses liver regeneration in chronic liver injury induced by HFD feeding.

A. Diagrams of liver-specific knockout (LKO) of SLC13A2 and study design. AAV-TBG-Cre-U6-Lacz (Ctrl) and AAV-TBG-Cre-U6-sgRNA (LKO) virus (5×10^{11} VP/mouse) were injected to the Rosa26-FSF-Cas9 knockin mouse via tail vein. The mice were subjected to HFD (60% kcal) for 20 weeks.

B. Serum ALT and AST levels.

C-D. Serum concentrations (C) and liver contents (D) of TG/TC.

E-F. Representative images of H&E staining (E) and ki67 IHC staining (F). Arrow heads are pointing to the double-nucleated hepatocytes.

G. Quantification for the ratios of Ki67-positive cells.

H. Representative images of sirius red staining.

I. Immunoblotting of proteins in signaling pathways and cholesterol synthesis.

Data are expressed as Mean \pm SEM (N=6, C-D and G; N=4, I). *P < 0.05 and **P < 0.01 LKO vs Ctrl group, two-tailed unpaired Student's *t* test.

2. What is the consequence of loss of SLC13A2 upon liver injury for a long term in terms of liver function and hepatocyte proliferation? Does the SLC13A2 deletion delay the regenerative process only for a short time followed by compensation from other pathways? Authors should have included data of a longer time point to show the liver function, liver mass, lipid metabolism, survival if there is a difference.

It is an insightful suggestion. We've analyzed liver function, liver mass, lipid metabolism and survival rate at 7d after PHx surgery. All the parameters except liver mass were not changed with the manipulation of SLC13A2 (Figure S4). It is understandable that the peak of liver metabolism remodeling and cell proliferation is between 1-3 days after the surgery. Also, survival rate is not changed after 7 days (Figure S4D, J) since the liver function and liver mass are almost fully recovered then. As the whole process of liver regeneration ends at 5 to 7 days after PHx along with the cessation of proliferation and restoration of hepatic architecture (Michalopoulos and DeFrances 1997), we don't think that there will be any difference regarding liver function, liver mass, lipid metabolism and survival rate at longer time than 7 d with the deletion of SLC13A2, which is a gene deregulated during the initiation and progression phase of liver regeneration. Further,

the application of chronic liver injury model induced by HFD confirmed the promotion role of SLC13A2 on liver regeneration for a long time (Figure 6). Based on the data, we conclude that SLC13A2-induced enhancement of hepatic regenerative capacity is effective for both acute and chronic liver injury.

Thanks for the suggestions! We have included the phenotypic data of mice at 7d after PHx into the supplementary data.

The new Figure S4 is shown as follows.

Figure S4 Limited effects of SLC13A2 at 7d after the surgery of PHx.

A-F. Liver regeneration was induced by PHx surgery with the overexpression of SLC13A2 and analysis was made after 7d. Body weight (A), blood glucose (B), iWAT weight (C), survival rate (D), RNA expression of genes for cholesterol synthesis, glucose metabolism and cell proliferation (E) and H&E staining (F) of GFP and SLC13A2-OE mice (N=6).

G-M. Liver regeneration was induced by PHx surgery with the liver-specific deletion of SLC13A2 and analysis was made after 7d. Body weight (G), blood glucose (H), iWAT weight (I), survival rate (J), Serum ALT/AST activity (K), RNA expression of genes for cholesterol synthesis, glucose metabolism and cell proliferation (L) and H&E staining (M) of Ctrl and SLC13A2-LKO mice (N=5).

Data are expressed as Mean \pm SEM.

3. Fig 3. What is the phenotype of SLC13A2 overexpression during homeostasis before PHx?

We've analyzed the role of SLC13A2 overexpression and knockout on the liver homeostasis before PHx.

When SLC13A2 was specifically knocked out in hepatocytes, no consistent differences were observed in the two liver-specific SLC13A2-KO (LKO) groups regarding blood glucose, body weight, liver weight, histological staining and expression of genes involving with glycolysis, TCA cycle and lipogenesis (Figure S2A-E). Further, we detected the activation of signaling pathway regulating cell proliferation and found that p-mTOR was slightly decreased while p-GSK3 β were significantly increased (Figure S2F). Similarly, when we overexpressed SLC13A2 in the liver, body weight, liver weight, blood glucose, histological morphology, gene expression and activation of signaling pathways were all comparable between the groups (Figure S3A-F). The data imply that hepatocyte-specific knockout or overexpression of SLC13A2 has limited effects on the liver homeostasis in quiescent liver without the insult of PHx surgery.

The new Figure S2-3 is shown as follows.

Supplementary figure S2

Figure S2 Effects of liver-specific SLC13A2 knockout in quiescent liver.

A-C. Body weight (A), liver weight (B) and blood glucose (C) of Ctrl and LKO mice.

D. qPCR analysis detected the expression of genes involving with glucose metabolism and lipid metabolism.

E. Representative H&E staining.

F. Immunoblots of total liver lysates.

Data are expressed as Mean \pm SEM (N=7).

Supplementary figure S3

Figure S3 Effects of liver-specific SLC13A2 overexpression without PHx surgery.

A-C. Body weight (A), liver weight (B) and blood glucose (C) of Ctrl and SLC13A2-OE mice (N=6).

D. RNA expression of genes involving with glucose, lipid and glutamine metabolism was analyzed by qPCR assay (N=4).

E. Representative liver histological staining.

F. Immunoblots of total liver lysates.

Data are expressed as Mean \pm SEM.

4. Fig 5. How does SLC13a2 regulate the cholesterol synthesis genes, the involve of mRNA expression level or the SLC13A2 transporter on cell membrane? For the overexpression study, authors should show the protein distribution in cells.

The reviewer is concerned about the mechanisms of SLC13A2 regulating cholesterol synthesis, which is an excellent point. By isolating cellular fractions with SLC13A2 overexpression, we proved that it is a membrane transporter (new Figure 8D). Using metabolomic analysis, we found that SLC13A2 transported citrate, instead of other well-known substrates (new Figure 8E and F). We also used ^{13}C -labeled citrate to verify the transporting substrate of SLC13A2 (new Figure 8G). The imported citrate was cleaved by ACLY and secreted acetyl-CoA which promoted the synthesis of cholesterol for liver regeneration (new Figure 9A-J). We also confirmed that citrate transported by SLC13A2 transcriptionally upregulates the expression of cholesterol synthesis genes, which was abolished with SLC13A2 deficiency (Figure 8H-I). With these improvements, we've strengthened the mechanistic basis of this study. Thank you very much for the good comments!

The new Figure 8 and 9 are shown as follows.

Figure 8

Figure 8. SLC13A2 promotes cell proliferation through the import of citrate for cholesterol synthesis as a membrane transporter in hepatocytes.

A. Cell viability (left) and colony formation capability (right) of AML12 cells after the infection with adenovirus, and cultured for designated days (cell viability) or 7 days (colony formation).

B. AML12 cells and primary hepatocytes were transduced with adenovirus for 48 h, and then stained with fluorescent dyes for cholesterol staining. Stars or arrow heads indicated cholesterol staining in hepatocytes or AML cells, respectively. Scale bar, 50 μm .

C. Cas9^{fl/fl} mice were injected with AAV virus (AAV-TBG-Cre-U6-LACZ, Ctrl; AAV-TBG-Cre-U6-SLC13A2 gRNA, LKO) through tail vein to ensure the liver-specific knockout of SLC13A2 in LKO group; primary hepatocytes were isolated, allowed adherence overnight, and cultured in the presence of vehicle or cholesterol (50 μM) for 48 h (left).

Primary hepatocytes were isolated from wild-type mice and then transduced with adenovirus (vector vs SLC13A2) for 48 h, followed by the treatment with vehicle or Lovastatin (10 μM) for 48 h (right).

D. Cellular fractions were isolated from primary hepatocytes with SLC13A2-overexpressing adenovirus transduction for 60 h, followed by immunoblotting for the cellular localization of SLC13A2 with specific controls.

E. Putative substrates for SLC13A2 transportation in the liver tissues from mice transduced with AAV at 1d after PHx surgery were measured using UPLC-MS analysis.

F. Primary hepatocytes were cultured and transduced with adenovirus for 60 h, then the putative substrates for SLC13A2 in both extracellular and intracellular compartment were analyzed using UPLC-MS analysis.

G. Primary hepatocytes with overexpression of SLC13A2 were treated with 125 μM [U-13C6] citrate for 1 h. The intracellular metabolites were then extracted for targeted metabolomics analysis of the m+6 citrate ratio.

H-I. Primary hepatocytes were isolated from Ctrl or LKO mice. After overnight culture for adherence, the cells were treated with 200 μM citrate for 24 h in DMEM medium with 2% LPDS. RT-qPCR and immunoblotting assays were used to analyze the expression of cholesterol-synthesis genes (H) and activation of cellular signaling pathway and cholesterol synthesis (I).

Data are expressed as Mean \pm SEM (N=3). *P < 0.05 and **P < 0.01 SLC13A2 vs Vector/GFP, LKO vs Ctrl, citrate vs vehicle; #P < 0.05 and ##P < 0.01 LKO+citrate vs Ctrl+citrate; two-tailed unpaired Student's t test (E-G), one-way (H) or two-way (A) ANOVA followed by Bonferroni multiple comparisons.

Figure 9

Figure 9. Citrate transported by SLC13A2 is catalyzed by ACLY, which secretes Acetyl-CoA for cholesterol synthesis to facilitate liver regeneration.

A. Diagrams of study design. Mice were injected with AAV virus (GFP: AAV-TBG-GFP; SLC13A2: AAV-TBG-SLC13A2 virus; 1×10^{11} VP/mouse) through tail vein to ensure the liver-specific overexpression of SLC13A2. ACLYi (20 mg/Kg) or vehicle were administered intragastrically once daily for consecutive 3 days. The surgery of PHx was performed on the third day and followed by 3 more doses of ACLYi administration. The mice were sacrificed for analysis at 3d after the surgery.

B. Liver weight versus body weight ratio.

C. Blood glucose.

D-E. Liver TG (D) and TC (E) content.

F. RNA expression of genes involving cholesterol synthesis and cell proliferation.

G. Representative images of H&E staining and BrdU incorporation. Scale bar, 100 μm .

H. Primary hepatocytes were cultured and transduced with adenovirus for 48 h, followed by the treatment with ACLYi in different concentrations (15, 20 and 25 μM) for 48 h.

I. AML12 cells were transduced with adenovirus for 48 h, followed by the treatment of ACLYi (25 μM) in the presence or absence of cholesterol (50 μM) for 48 h. EdU incorporation assay was performed with the addition of EdU for 2 h. Scale bar, 50 μm .

J. Cholesterol staining using filipin fluorescence dye were performed in AML12 cells after adenovirus transduction for 48 h followed by the treatment of ACLYi (25 μM), in the absence or presence of cholesterol (50 μM) for 48 h. Scale bar, 50 μm .

Data are expressed as Mean \pm SEM (N=3). *P < 0.05 and **P < 0.01 SLC13A2 vs GFP, ACLYi vs vehicle (H); #P < 0.05 and ##P < 0.01 ACLYi vs vehicle (B-F); one-way ANOVA followed by Bonferroni multiple comparisons.

5. Figure legend and method sections need more details about the experiments, cell types and treatment. Fig7b staining need to be marked clearly in the figure, and Fig7a standard deviation is missing.

We have added details of experimental design including cell types and treatment in the figure legends and methodology sections. Also, marks of stars or arrow heads have

been used to indicate cholesterol staining in hepatocytes or AML cells respectively in new Figure 8B. Standard deviation has been added in new Figure 8A.

The new Figure 8A and 8B are shown as follows.

6. Fig 6H and E, G, lovastatin treatment showed increased cell proliferation, p-JNK pathway, and FL-Srebp2 in H but not in E and G.

Sorry for the misleading information in the previous Figure 6H (new Figure 7H). The treatment of lovastatin without the overexpression of SLC13A2 doesn't increase cell proliferation, as P-JNK, FL-SREBP2 and PCNA wasn't increased when we reanalyzed the normalized quantification results. We have replaced with more representative blots.

The new Figure 7H is shown below.

7. Fig 6. how does the data look like in 7 days after PHx? 3 days has no obvious difference of LW/BW ratio as shown in 7 days

We compared the treatment of lovastatin with vehicle in the mice with SLC13A2 overexpression at 7d post PHx surgery (Figure S6A). There is no difference regarding liver/body weight (Figure S6B), blood glucose (Figure S6C), RNA expression of cholesterol synthesis and proliferating genes (Figure S6D), liver TG and serum TG/TC, and histological staining, while liver TC content was significantly increased with the treatment of lovastatin before the PHx surgery (Figure S6E). The reason for this phenotype might be due to the short elimination half-life of lovastatin, which is about 1-3 h (Schachter 2005). When we design this experiment, we don't want to suppress the endogenous cholesterol synthesis extensively, but only the cholesterol synthesis induced by SLC13A2 overexpression. We can see the decreased BrdU incorporation by lovastatin treatment in SLC13A2-OE group (Figure 7D-E, S6F), which is a gold standard for liver regeneration. It is worthy trying different dose regime of lovastatin treatment to show significantly suppressed liver/body weight ratio in the following studies.

Thanks for the good comments! We have included the 7d data into the supplementary results.

The new Figure S6 is shown as follows.

Figure S6

Figure S6. The effect of lovastatin on SLC13A2-promoted liver regeneration at 7d after PHx surgery.

A. Diagrams of study design. Mice were injected with AAV virus (GFP: AAV-TBG-GFP; SLC13A2: AAV-TBG-SLC13A2 virus; 1×10^{11} VP/mouse) through tail vein to ensure the liver-specific overexpression of SLC13A2, followed by the intragastrical administration of Lovastatin (4 mg/Kg) or vehicle once a day for consecutive 3 days. Then the surgery of PHx was operated, and the mice were sacrificed after 7 days.

B. Liver weight versus body weight ratio.

C. Blood glucose.

D. RNA expression of genes involving cholesterol synthesis and cell proliferation.

E. Liver contents and serum concentrations of TG/TC.

F. Representative images of H&E staining and BrdU incorporation.

Data are expressed as Mean \pm SEM (N=5-6). *P < 0.05 Lovastatin vs Vehicle group, two-tailed unpaired Student's *t* test.

8. Fig 7C,D, treatment of cholesterol upon KO should be blot together with the control group. Like this, authors can evaluate the contribution of the cholesterol treatment compared to the control group. Specifically, the Upper and bottom panels should be blot together to check the contribution of cholesterol in proliferation and signaling pathways listed.

When we repeated the experiment and blotted the groups together, the cholesterol or lovastatin treatment group was able to be compared with the control group. The attenuated expression of cholesterol synthesis genes and cell proliferation signaling by SLC13A2 deficiency was nearly completely rescued by the treatment of cholesterol (new Figure 8C left). Similarly, the elevated expression of genes and signaling pathway by SLC13A2 overexpression was almost fully abolished with the treatment of lovastatin (new Figure 8C right). We have replaced the old figures with the newly generated ones. Thanks for the good comment!

New Figure 8C is shown below.

9. How is the *in vivo* phenotype of ACLYi? This is to evaluate if citrate transportation is a key mediator of SLC13A2 mediated liver regeneration.

We have further proved the effects of ACLYi in SLC13A2-promoted liver regeneration in SLC13A2-OE mouse model (Figure 9A). After sacrifice of the mice, cell proliferation, serum and liver TG/TC have been analyzed. It was delineated that the application of ACLYi *in vivo* suppressed SLC13A2-boosted liver regeneration dependent upon cholesterol synthesis as shown with increased liver TG/TC and expression of genes for cholesterol synthesis and cell proliferation (Figure 9B-D). The results confirm that citrate transportation is a key mediator for SLC13A2-boosted liver regeneration.

The new 9 is shown as follows.

Figure 9. Citrate transported by SLC13A2 is catalyzed by ACLY, which secretes Acetyl-CoA for cholesterol synthesis to facilitate liver regeneration.

A. Diagrams of study design. Mice were injected with AAV virus (GFP: AAV-TBG-GFP; SLC13A2: AAV-TBG-SLC13A2 virus (1×10^{11} VP/mouse) through tail vein to ensure the

liver-specific overexpression of SLC13A2. ACLYi (20 mg/Kg) or vehicle were administered intragastrically once daily for consecutive 3 days. The surgery of PHx was performed on the third day and followed by 3 more doses of ACLYi administration. The mice were sacrificed for analysis at 3d after the surgery.

B. Liver weight versus body weight ratio.

C. Blood glucose.

D-E. Liver TG (D) and TC (E) content.

F. RNA expression of genes involving cholesterol synthesis and cell proliferation.

G. Representative images of H&E staining and BrdU incorporation. Scale bar, 100 μ m.

H. Primary hepatocytes were cultured and transduced with adenovirus for 48 h, followed by the treatment with ACLYi in different concentrations (15, 20 and 25 μ M) for 48 h.

I. AML12 cells were transduced with adenovirus for 48 h, followed by the treatment of ACLYi (25 μ M) in the presence or absence of cholesterol (50 μ M) for 48 h. EdU

incorporation assay was performed with the addition of EdU for 2 h. Scale bar, 50 μ m.

J. Cholesterol staining using filipin fluorescence dye were performed in AML12 cells after adenovirus transduction for 48 h followed by the treatment of ACLYi (25 μ M), in the absence or presence of cholesterol (50 μ M) for 48 h. Scale bar, 50 μ m.

Data are expressed as Mean \pm SEM (N=3). *P < 0.05 and **P < 0.01 SLC13A2 vs GFP, ACLYi vs vehicle (H); #P < 0.05 and ##P < 0.01 ACLYi vs vehicle (B-F); one-way ANOVA followed by Bonferroni multiple comparisons.

10. Along the same line as 9, Authors should check the contribution of Citrate *in vitro* and *in vivo* in terms of liver regeneration. Knockout SLC13A2 and treat with citrate in hepatocytes and check if SLC13A2 is the major transporter for citrate during liver regeneration.

The application of ACLYi *in vivo* and *in vitro* confirms the contribution of citrate to SLC13A2-promoted liver regeneration (Figure 9).

Also, as a part for the response to comment #2 from this referee, we supplemented citrate into the WT and KO hepatocytes and detected the expression of cholesterol synthesis genes as well as the activation of cellular signaling for cell proliferation. It shows that citrate increases the expression of cholesterol synthesis genes and cell signaling pathway for proliferation, which is compromised by the knockout of SLC13A2 in hepatocytes (Figure 8H-I).

With these data, we are confident to conclude that SLC13A2 is the major transporter for citrate in order to synthesize cholesterol for liver regeneration.

Thanks a lot for the good comment!

New Figure 8H-I

H-I. Primary hepatocytes were isolated from Ctrl or LKO mice. After overnight culture for adherence, the cells were treated with 200 μ M citrate for 24 h in DMEM medium with 2% LPDS. RT-qPCR and immunoblotting assays were utilized to analyze the expression of cholesterol synthesis genes (H) as well as the activation of cellular signaling pathways and cholesterol synthesis (I). Data are expressed as Mean \pm SEM (N=3). *P < 0.05 and **P < 0.01 SLC13A2 vs Vector/GFP, LKO vs Ctrl, citrate vs vehicle; #P < 0.05 and ##P < 0.01 LKO+citrate vs Ctrl+citrate; two-tailed unpaired Student's t test (E-G), one-way (H) or two-way (A) ANOVA followed by Bonferroni multiple comparisons.

Comments from Referee #2:

Hepatocytes undergo metabolism remodeling during liver regeneration, in this manuscript Shi and colleagues illustrated significance of cholesterol metabolism mediated by SLC13A2 for liver regeneration. The authors demonstrated SLC13A2 promote liver regeneration after partial hepatectomy (PHx) with knockout and overexpression mouse model induced by Adeno-Associated Virus (AAV) transfection. Mechanically, SLC13A2 encode multiple transmembrane transporters, which transport citrate into intracellular for de novo cholesterol synthesis and finally promote liver regeneration.

We thank this reviewer for his/her positive comments on our manuscript.

Major points:

1) In Figure 1A, the authors analyzed transcriptomic profiles from other papers and SLC13A2 was choose for the candidate genes from differently expressed genes during liver regeneration. But different time points are on behalf of different stage of liver regeneration, the author should introduce the screen strategy in detail, it is best to provide gene list of every part of Venn diagram.

We chose the datasets which include time points representing the initiation and progression of liver regeneration (Figure 1A-B). Time points chosen have been stated clearly to introduce the screen strategy in detail. The gene lists of every part of Venn diagram have been provided as Expanded View Content of Figure 1A.

New Figure 1A-B is shown below.

A. Analysis of differentially expressed genes during liver regeneration induced by partial hepatectomy (PHx) in GEO datasets.

B. The expression of *Slc13a2* in the above GEO datasets.

2) The authors should explain the function of EGF treatment for primary hepatocyte in Figure 5C.

The treatment of EGF is used to partially mimic the essential humoral regulators for hepatocellular proliferation faced by hepatocytes during liver regeneration *in vitro* (Kang, Mars, and Michalopoulos 2012). EGF, as a mitogen for liver regeneration, is able to stimulate hepatocytes DNA synthesis and cell proliferation in culture (Block et al. 1996). So, it is a good option for the usage *in vitro* on hepatocytes.

The reason for this experiment has been included in revision. Thanks for the good comments!

3) There is a feedback regulation in cholesterol metabolism. Enough cholesterol inhibits SREBP2 hydrolysis, then reduce nSREBP2 level and transcript of Hmgcr. But in Figure 7C (bottom left), why cholesterol supplement result in the rise of HMGCRC and nSREBP2?

We agree that persistently enough cholesterol inhibits SREBP2 cleavage and reduces nSREBP2 and HMGCR expression in a feed-back manner. In Figure 7C, when the hepatocytes were knocked out with SLC13A2, the cholesterol synthesis was suppressed in these cells. In a setting of insufficient cholesterol supply, the application of cholesterol increases SREBP2 cleavage, nSREBP2, HMGCR and LDL expression. When cholesterol abundance is low, mTORC1 localizes in the cytosol and remains inactive. When intracellular cholesterol concentration increases, mTORC1 translocates to the lysosomal membrane and activates its kinase function, which initiates anabolic pathway for cell growth(Lim et al. 2019). Activation of mTORC1 triggers SREBP2 cleavage and translocation (Eid et al. 2017), phosphorylates USP20 and subsequently stabilizes HMGCR at the fasting-feeding transition (Lu et al. 2020). This fasting condition reflects insufficient cholesterol supply, while refeeding means regained supply of cholesterol, which is very consistent with our settings. Thanks for the good comments, we discussed the results by focusing on the disparate settings with previous dogma.

4) SLC13A2 play an essential role in initiation of liver regeneration, the authors should discuss that why the SLC13A2 down-regulated after PHx in Figure 1.

SLC13A2 is decreased as a rapid response to the initial signal triggering liver regeneration. Based on our data, we conclude that SLC13A2 is important for the progression of liver regeneration. It is increased along with the recovery of liver mass and function by providing essential metabolites for the synthesis of cholesterol during liver regeneration.

When we dissected the complicated insults in the early stage of PHx surgery for the initiation of liver regeneration, we tried to apply combined fatty acid (PAOA), inflammatory cytokines (TNF α) and glucose deprivation to primary hepatocytes. We found that combined fatty acid significantly decreased the transcriptional regulation of SLC13A2, instead of other insults, which was in accordance with the suppressed expression of SLC13A2 by HFD feeding (GSE68360). In addition, some transcription factors might get involved in the transcriptional regulation of SLC13A2 during liver

regeneration, including HNF4 α (GSE210842), which is responded to signals of fatty acid and inflammation. In our preliminary study, we also confirmed that the overexpression of HNF4 α increased the transcription of SLC13A2. Previous study reveals a rapid decline in nuclear and cytoplasmic HNF4 α protein levels, along with decreased target gene expression, within 1h after PHx and recovered after 3d. Also, hepatocyte-specific deletion of HNF4 α resulted in 100% mortality of this surgery (Huck et al. 2019). What's the role of SLC13A2 in HNF4 α -involved maintenance of liver regenerative capacity deserves full address in the future. Further elucidation on the regulatory mechanisms of SLC13A2 expression in response to the liver injury signals will provide insightful suggestions for drug discovery strategies targeting this transporter.

We have added discussion on the possible reasons why SLC13A2 is downregulated in the initiation phase of PHx. Since these data are still preliminary, they deserve further elucidation in the future. We prefer not to include them in this paper.

Thank you very much for the good comment!

Our preliminary data are shown below.

5) In Figure 3E, overexpression of SLC13A2 reduce the blood glucose level after PHx, but in Figure 6F, SLC13A2 increase the blood glucose. Meanwhile, result description in manuscript of Figure 3E (Liver/body weight ratio was significantly increased, while blood glucose at 1 day after the surgery was decreased with the overexpression of SLC13A2, indicating accelerated restoration of liver mass and functions by SLC13A2) and 6F (The increase of blood glucose reflecting the restoration of liver function was also induced by

SLC13A2, which was suppressed by the pretreatment of lovastatin) is also opposite.

Thank you very much for your careful reading of our data! Figure 3E and previous Figure 6F (new Figures 7F and 9C) showing blood glucose are different time points after PHx surgery. At 1d post PHx, overexpression of SLC13A2 repeatedly decreases blood glucose (Figure 3E). While at 3d post PHx, the blood glucose increases with the overexpression of SLC13A2 (new Figures 7F and 9C). We think the reason for this disparate result is that blood glucose is supposed to be dynamically regulated catering to the requirement of liver regeneration. In the initiation of liver regeneration (1d), decrease of blood glucose acts as an initial signal to trigger the start of liver regeneration, as reported previously that glucose supplementation suppresses hepatectomy-induced hepatocellular proliferation (Huang and Rudnick 2014; Weymann et al. 2009; Holecek 1999). After the initiation phase, metabolic adaptation occurs including suppression of liver glycolytic activity and shift of pyruvate metabolism, to obtain the maintenance of glucose homeostasis. In the propagation phase of liver regeneration, blood glucose acts as a predominant energy source for cell proliferation, which has huge energetic demand, as evidenced that prolonged hypoglycemia impaired liver regeneration and reduced mice survival after PHx (Alvarez-Guaita et al. 2020). Therefore, both decrease at 1d and increase at 3d of blood glucose induced by SLC13A2 overexpression should be beneficial for liver regeneration. We have added discussion on these results, thanks for the insightful suggestions!

Minor Points:

1) The authors should provide the specific time point of different transcriptome profile in Figure 1B.

The specific time point of different transcriptome profile have been provided in Figure 1B.

2) In Figure 1G, it seemed no significant change in PCNA after PHx, especially in PHx3d which is the proliferative stage of liver regeneration. The author should check it

again.

PCNA is truly increased in PHx 1d and 3d, we repeated that for many times and have more representative blots to replace the original one.

3) It is recommended that changing HSP90 (Top left) to ACTIN in Figure 4B to keep the consistency of reference gene in one figure.

We have more than one internal control when doing the analysis, ACTIN has replaced HSP90 to keep consistency.

4) The authors showed metabolomic data in primary hepatocytes after SLC13A2 overexpression, but no further exploration in the following works. The authors should explain more reason of this metabolomic data in manuscript.

The metabolomic data are important for us to conceal the mechanisms for SLC13A2 in promoting liver regeneration. In the following work, we focus on citrate which is based on the metabolomic data, including how it is used to synthesize cholesterol and promote hepatocellular proliferation. In revision, we have provided more evidence to support the mechanisms of SLC13A2-transported citrate in the promotion of liver regeneration, and have also made the logic flow more fluently. Thanks for the constructive comment, we improved the importance of metabolomic data which is the key of this manuscript.

We appreciate for Editors/ Reviewers' work earnestly, and hope that the correction will meet the approval.

Once again, thank you very much for your comments and suggestions.

Sincerely,

Jing Xiong, Ph.D.

Professor

Department of Pharmacology

School of Pharmacy

China Pharmaceutical University

E-mail: jxiong@cpu.edu.cn

Haiping Hao, Ph.D.

Professor

Jiangsu Provincial Key Laboratory of Drug Metabolism and Pharmacokinetics

State Key Laboratory of Natural Medicines

China Pharmaceutical University

E-mail address: haipinghao@cpu.edu.cn

References:

- Alvarez-Guaita, A., P. Blanco-Munoz, E. Meneses-Salas, M. Wahba, A. H. Pollock, J. Jose, M. Casado, M. Bosch, R. Artuch, K. Gaus, A. Lu, A. Pol, F. Tebar, S. E. Moss, T. Grewal, C. Enrich, and C. Rentero. 2020. 'Annexin A6 Is Critical to Maintain Glucose Homeostasis and Survival During Liver Regeneration in Mice', *Hepatology*, 72: 2149-64.
- Block, G. D., J. Locker, W. C. Bowen, B. E. Petersen, S. Katyal, S. C. Strom, T. Riley, T. A. Howard, and G. K. Michalopoulos. 1996. 'Population expansion, clonal growth, and specific differentiation patterns in primary cultures of hepatocytes induced by HGF/SF, EGF and TGF alpha in a chemically defined (HGM) medium', *J Cell Biol*, 132: 1133-49.
- Eid, W., K. Dauner, K. C. Courtney, A. Gagnon, R. J. Parks, A. Sorisky, and X. Zha. 2017. 'mTORC1 activates SREBP-2 by suppressing cholesterol trafficking to lysosomes in mammalian cells', *Proc Natl Acad Sci U S A*, 114: 7999-8004.
- Haideri, S. S., A. C. McKinnon, A. H. Taylor, P. Kirkwood, P. J. Starkey Lewis, E. O'Duibhir, B. Vernay, S. Forbes, and L. M. Forrester. 2017. 'Injection of embryonic stem cell derived macrophages ameliorates fibrosis in a murine model of liver injury', *NPJ Regen Med*, 2: 14.

- Holecek, M. 1999. 'Nutritional modulation of liver regeneration by carbohydrates, lipids, and amino acids: a review', *Nutrition*, 15: 784-8.
- Huang, J., and D. A. Rudnick. 2014. 'Elucidating the metabolic regulation of liver regeneration', *Am J Pathol*, 184: 309-21.
- Huck, I., S. Gunewardena, R. Espanol-Suner, H. Willenbring, and U. Apte. 2019. 'Hepatocyte Nuclear Factor 4 Alpha Activation Is Essential for Termination of Liver Regeneration in Mice', *Hepatology*, 70: 666-81.
- Kang, L. I., W. M. Mars, and G. K. Michalopoulos. 2012. 'Signals and cells involved in regulating liver regeneration', *Cells*, 1: 1261-92.
- Lim, C. Y., O. B. Davis, H. R. Shin, J. Zhang, C. A. Berdan, X. Jiang, J. L. Counihan, D. S. Ory, D. K. Nomura, and R. Zoncu. 2019. 'ER-lysosome contacts enable cholesterol sensing by mTORC1 and drive aberrant growth signalling in Niemann-Pick type C', *Nat Cell Biol*, 21: 1206-18.
- Lu, X. Y., X. J. Shi, A. Hu, J. Q. Wang, Y. Ding, W. Jiang, M. Sun, X. Zhao, J. Luo, W. Qi, and B. L. Song. 2020. 'Feeding induces cholesterol biosynthesis via the mTORC1-USP20-HMGCR axis', *Nature*, 588: 479-84.
- Michalopoulos, G. K., and M. C. DeFrances. 1997. 'Liver regeneration', *Science*, 276: 60-6.
- Schachter, M. 2005. 'Chemical, pharmacokinetic and pharmacodynamic properties of statins: an update', *Fundam Clin Pharmacol*, 19: 117-25.
- Weymann, A., E. Hartman, V. Gazit, C. Wang, M. Glauber, Y. Turmelle, and D. A. Rudnick. 2009. 'p21 is required for dextrose-mediated inhibition of mouse liver regeneration', *Hepatology*, 50: 207-15.

Dear Dr Xiong, dear Dr Hao,

Thank you for submitting your revised manuscript (EMBOJ-2024-118432R) to The EMBO Journal, as well for your patience with our response. Your amended study was sent back to the two referees for their scientific evaluation, and we have received re-comments from both of them, which I enclose below. As you will see, the experts state that the work has been substantially enhanced by the revisions and they are now in favour of publication, pending minor revision.

Thus, we are pleased to inform you that your manuscript has been accepted in principle for publication in The EMBO Journal.

Please consider the remaining minor issues by referee #1 carefully and adjust data presentation where appropriate.

We also now need you to take care of a number of issues related to formatting and data presentation as detailed below, which should be addressed at resubmission.

Please contact me at any time if you have additional questions related to below points.

As you might have seen on our web page, every paper at the EMBO Journal now includes a 'Synopsis', displayed on the html and freely accessible to all readers. The synopsis includes a 'model' figure as well as 2-5 one-short-sentence bullet points that summarize the article. I would appreciate if you could provide this figure and the bullet points.

Thank you for giving us the chance to consider your manuscript for The EMBO Journal. I look forward to your final revision.

Again, please contact me at any time if you need any help or have further questions.

Kind regards,

Daniel Klimmeck

>> Remove the figures from the manuscript .doc file.

>> Limit the abstract to maximally 175 words.

>> Author Contributions: Please remove the author contributions information from the manuscript text. Note that CRediT has replaced the traditional author contributions section as of now because it offers a systematic machine-readable author contributions format that allows for more effective research assessment. and use the free text boxes beneath each contributing author's name to add specific details on the author's contribution.

More information is available in our guide to authors.
<https://www.embopress.org/page/journal/14602075/authorguide>

>> Adjust the title of the 'Declaration of Interests' section to 'Disclosure and Competing Interests Statement' and move after Acknowledgements.

>> Please provide a completed Author Checklist.

>> Correct order of manuscript sections: Abstract / Keywords / Introduction / Results / Discussion / Methods / Data Availability / Acknowledgements / Disclosure and competing interests statement // References / Figure legends / Tables and their legends / Expanded View Figure legends

>> Callouts: add figure callouts in the manuscript text for Fig 9B and Table S3.

>> Figures in separate files: Figures should be removed from the manuscript text and uploaded as individual, high resolution figure files. Legends should be placed after the References.

>> Appendix file with ToC: The file with supplementary figures should be labelled "Appendix" and uploaded in PDF format. It should have a table of contents, including page numbers, added to the first page. The figures should be renamed "Appendix Figure A 1" etc. The duplicate TIF files for these figures can be removed. Tables S1-3 should be renamed "Appendix Table S1" etc and they should be added to the appendix PDF.

>> References: adjust reference format to EMBO Journal format, 10 authors et al, and place References after the Discussion, before figure legends.

>> Add a Reagents and Tools table to the Methods section, as a separate file using the existing template in the Guide For Authors, listing key reagents, experimental models, software and relevant equipment.

>> Data not shown: remove the comment on p. 22 or add respective data to the manuscript.

>> The excel table labelled as "Figure 1A Expanded View Content" should be turned into source data.

>> Consider additional changes and comments from our production team as indicated below:

- Figure legends:

1. Please note that the exact p values are not provided in the legends of figures 1b, d-f, h; 2c-e, h; 3c-e, h; 4a-b; 5b, d; 6b-d, g, i; 7b-c, e-h; 8a, c, e-h; 9b-f, h.
2. Please note that information related to n is missing in the legend of figure 1b.
3. Although 'n' is provided, please describe the nature of entity for 'n' in the legends of figures 1d-f, h; 2c-e, h; 3c-e, h; 4a; 5b; 6b-d, g, i; 7b-c, e-g; 8a, f-g; 9b-f, h.

Referee #1:

Thank the authors to address my questions with substantial amount of data during the revision. I have no major concerns. Authors should go through the manuscript and make sure the detailed information are intact, including scale bars(such as Fig.9) and labels(such as Fig6A). thank you and congrats to the authors.

Referee #2:

No further comments.

The authors addressed the remaining editorial issues.

Professor

Department of Pharmacology

School of Pharmacy

China Pharmaceutical University

E-mail: jxiong@cpu.edu.cn

Haiping Hao, Ph.D.

Professor

Jiangsu Provincial Key Laboratory of Drug Metabolism and Pharmacokinetics

State Key Laboratory of Natural Medicines

China Pharmaceutical University

E-mail address: haipinghao@cpu.edu.cn

References:

- Alvarez-Guaita, A., P. Blanco-Munoz, E. Meneses-Salas, M. Wahba, A. H. Pollock, J. Jose, M. Casado, M. Bosch, R. Artuch, K. Gaus, A. Lu, A. Pol, F. Tebar, S. E. Moss, T. Grewal, C. Enrich, and C. Rentero. 2020. 'Annexin A6 Is Critical to Maintain Glucose Homeostasis and Survival During Liver Regeneration in Mice', *Hepatology*, 72: 2149-64.
- Block, G. D., J. Locker, W. C. Bowen, B. E. Petersen, S. Katyal, S. C. Strom, T. Riley, T. A. Howard, and G. K. Michalopoulos. 1996. 'Population expansion, clonal growth, and specific differentiation patterns in primary cultures of hepatocytes induced by HGF/SF, EGF and TGF alpha in a chemically defined (HGM) medium', *J Cell Biol*, 132: 1133-49.
- Eid, W., K. Dauner, K. C. Courtney, A. Gagnon, R. J. Parks, A. Sorisky, and X. Zha. 2017. 'mTORC1 activates SREBP-2 by suppressing cholesterol trafficking to lysosomes in mammalian cells', *Proc Natl Acad Sci U S A*, 114: 7999-8004.
- Haideri, S. S., A. C. McKinnon, A. H. Taylor, P. Kirkwood, P. J. Starkey Lewis, E. O'Duibhir, B. Vernay, S. Forbes, and L. M. Forrester. 2017. 'Injection of embryonic stem cell derived macrophages ameliorates fibrosis in a murine model of liver injury', *NPJ Regen Med*, 2: 14.

- Holecek, M. 1999. 'Nutritional modulation of liver regeneration by carbohydrates, lipids, and amino acids: a review', *Nutrition*, 15: 784-8.
- Huang, J., and D. A. Rudnick. 2014. 'Elucidating the metabolic regulation of liver regeneration', *Am J Pathol*, 184: 309-21.
- Huck, I., S. Gunewardena, R. Espanol-Suner, H. Willenbring, and U. Apte. 2019. 'Hepatocyte Nuclear Factor 4 Alpha Activation Is Essential for Termination of Liver Regeneration in Mice', *Hepatology*, 70: 666-81.
- Kang, L. I., W. M. Mars, and G. K. Michalopoulos. 2012. 'Signals and cells involved in regulating liver regeneration', *Cells*, 1: 1261-92.
- Lim, C. Y., O. B. Davis, H. R. Shin, J. Zhang, C. A. Berdan, X. Jiang, J. L. Coughlin, D. S. Ory, D. K. Nomura, and R. Zoncu. 2019. 'ER-lysosome contacts enable cholesterol sensing by mTORC1 and drive aberrant growth signalling in Niemann-Pick type C', *Nat Cell Biol*, 21: 1206-18.
- Lu, X. Y., X. J. Shi, A. Hu, J. Q. Wang, Y. Ding, W. Jiang, M. Sun, X. Zhao, J. Luo, W. Qi, and B. L. Song. 2020. 'Feeding induces cholesterol biosynthesis via the mTORC1-USP20-HMGCR axis', *Nature*, 588: 479-84.
- Michalopoulos, G. K., and M. C. DeFrances. 1997. 'Liver regeneration', *Science*, 276: 60-6.
- Schachter, M. 2005. 'Chemical, pharmacokinetic and pharmacodynamic properties of statins: an update', *Fundam Clin Pharmacol*, 19: 117-25.
- Weymann, A., E. Hartman, V. Gazit, C. Wang, M. Glauber, Y. Turmelle, and D. A. Rudnick. 2009. 'p21 is required for dextrose-mediated inhibition of mouse liver regeneration', *Hepatology*, 50: 207-15.

Dear Dr Xiong,

Thank you for submitting the revised version of your manuscript. I have now evaluated your amended manuscript and concluded that the remaining minor concerns have been sufficiently addressed.

I am thus pleased to inform you that your manuscript has been accepted for publication in the EMBO Journal.

On a different note, I would like to alert you that EMBO Press offers a format for a video-synopsis of work published with us, which essentially is a short, author-generated film explaining the core findings in hand drawings, and, as we believe, can be very useful to increase visibility of the work. Please see the following link for representative examples and their integration into the article web page:

<https://www.embopress.org/doi/full/10.15252/emboj.2019103932>

Best regards,

Daniel Klimmeck

Daniel Klimmeck, PhD
Senior Editor
The EMBO Journal
EMBO
Postfach 1022-40
Meyershofstrasse 1
D-69117 Heidelberg
contact@embojournal.org
Submit at: <http://emboj.msubmit.net>